



# Assessment of upper tropospheric and stratospheric water vapour and ozone in reanalyses as part of S-RIP

Sean M. Davis[1,2], Michaela I. Hegglin[3], Masatomo Fujiwara[4], Rossana Dragani[5], Yayoi Harada[6], Chiaki Kobayashi[6,7], Craig Long[8], Gloria L. Manney[9,10], Eric Nash[11], Gerald L. Potter[12], Susann Tegtmeier[13], Tao Wang[14], Krzysztof Wargan[11,15], Jonathon S. Wright[16]

[1]Earth System Research Laboratory, National Oceanic and Atmospheric Administration, Boulder, CO 80305, USA
[2]Cooperative Institute for Research in Environmental Sciences (CIRES), University of Colorado at Boulder, Boulder, CO 80309, USA
[3]Department of Meteorology, University of Reading, Reading, RG6 6BX, UK
[4]Faculty of Environmental Earth Science, Hokkaido University, Sapporo 060-0810, Japan
[5]European Centre for Medium-Range Weather Forecasts, Reading, RG2 9AX, UK
[6]Japan Meteorological Agency, Tokyo, 100-8122, Japan
[7]Climate Research Department, Meteorological Research Institute, JMA, Tsukuba, 305-0052, Japan
[8]Climate Prediction Center, National Centers for Environmental Prediction, National Oceanic and Atmospheric Administration, College Park, MD 20740, USA
[9]NorthWest Research Associates, Socorro, NM 87801, USA
[10]Department of Physics, New Mexico Institute of Mining and Technology, Socorro, NM 87801, USA
[11]Science Systems and Applications, Inc., Lanham, Maryland 20706, USA
[12]NASA Center for Climate Simulation, Code 606.2, NASA Goddard Space Flight Center, Greenbelt MD 20771, USA
[13]GEOMAR Helmholtz Centre for Ocean Research Kiel, Kiel, 24105, Germany
[14]NASA Jet Propulsion Laboratory/California Institute of Technology, Pasadena, CA 91109, USA
[15]Global Modeling and Assimilation Office, Code 610.1, NASA Goddard Space Flight Center, Greenbelt, MD 20771, USA
[16]Department of Earth System Science, Tsinghua University, Beijing, 100084, China

Correspondence to: Sean M. Davis (sean.m.davis@noaa.gov)




**Abstract.** Reanalysis data sets are widely used to understand atmospheric processes and past variability, and are often used to stand in as "observations" for comparisons with climate model output. Because of the central role of water vapour (WV) and ozone ($O_3$) in climate change, it is important to understand how accurately and consistently these species are represented in existing global reanalyses. In this paper, we present the results of WV and $O_3$ intercomparisons that have been performed

as part of the SPARC (Stratosphere-troposphere Processes and their Role in Climate) Reanalysis Intercomparison Project (S-RIP). The comparisons cover a range of timescales and evaluate both inter-reanalysis and observation-reanalysis differences. We also provide a systematic documentation of the treatment of WV and $O_3$ in current reanalyses to aid future research and guide the interpretation of differences amongst reanalysis fields.

The assimilation of total column ozone (TCO) observations in newer reanalyses results in realistic representations of TCO in

reanalyses except when data coverage is lacking, such as during polar night. The vertical distribution of ozone is also relatively well represented in the stratosphere in reanalyses, particularly given the relatively weak constraints on ozone vertical structure provided by most assimilated observations and the simplistic representations of ozone photochemical processes in most of the reanalysis forecast models. However, significant biases in the vertical distribution of ozone are found in the upper troposphere and lower stratosphere in all reanalyses.

In contrast to $O_3$, reanalysis estimates of stratospheric WV are not directly constrained by assimilated data. Observations of atmospheric humidity are typically used only in the troposphere, below a specified vertical level at or near the tropopause. The fidelity of reanalysis stratospheric WV products is therefore mainly dependent on the reanalyses' representation of the physical drivers that influence stratospheric WV, such as temperatures in the tropical tropopause layer, methane oxidation, and the stratospheric overturning circulation. The lack of assimilated observations and known deficiencies in the

representation of stratospheric transport in reanalyses result in much poorer agreement amongst observational and reanalysis estimates of stratospheric WV. Hence, stratospheric WV products from the current generation of reanalyses should generally not be used in scientific studies.

## 1 Introduction

Ozone and water vapour are trace gases of fundamental importance to the radiative budget of the stratosphere. Because of

their impact on stratospheric temperatures, winds, and the circulation (e.g., Dee et al., 2011), ozone and water vapour are represented as prognostic variables in almost all current reanalysis systems. However, the degree of sophistication to which ozone and water vapour fields and their variability are represented depends on the reanalysis system, which observations it assimilates, which microphysical and chemical parameterizations it includes, and how those parameterizations affect the trace gas distributions. The accuracy and consistency of analysis and reanalysis ozone and water vapour fields in the upper

troposphere and stratosphere has only been addressed for a limited subset of diagnostics and analysis/reanalysis systems by a few studies (e.g., Dessler and Davis, 2010; Jiang et al., 2015; Geer et al., 2006; Thornton et al., 2009).





As part of the SPARC (Stratosphere-troposphere Processes and their Role in Climate) Reanalysis Intercomparison Project (S-RIP), we conducted the first comprehensive assessment of how realistically and consistently reanalyses represent water vapour and ozone in the upper troposphere and stratosphere. In particular, the goals of this paper are to (1) provide a comprehensive overview of how ozone and water vapour are treated in reanalyses, (2) evaluate the accuracy of ozone and

water vapour in reanalyses against both assimilated and independent (non-assimilated) observations, and (3) provide guidance to the community regarding the proper usage and limitations of reanalysis ozone and water vapour fields in the upper troposphere and stratosphere.

**2 Description of ozone and water vapour in reanalyses**

In this section, we provide information on how ozone and water vapour are represented in reanalyses. The information

compiled here expands on that provided by Fujiwara et al. (2017), who presented a comprehensive overview of the reanalysis systems and their assimilated observations, including a basic discussion of the treatment of ozone and water vapour.

In most reanalyses, ozone and water vapour are prognostic variables that are affected by the assimilated observations (see Tables 1 and 2 for an overview of key aspects of these fields). The assimilated observations affecting the water vapour fields

in reanalyses include some combination of radiosonde humidity profiles, GNSS-RO bending angles, and either radiances or retrievals from satellite microwave and infrared sounders such as TOVS, ATOVS, and SSM/I (see Appendix A for a list of all abbreviations). These observational data affect the reanalysis water vapour fields in the lower atmosphere, but radiosonde humidity data are not assimilated above a specified level in the upper troposphere (typically between 300 hPa and 100 hPa, see Table 2). Even though radiosonde humidity data may not be assimilated above a certain level, analysis increments are

possible at higher levels unless the vertical correlations of the background errors are set to zero. Where relevant, this cutoff level above which analysis increments are disallowed has been noted in Table 2.

Because stratospheric water vapour data are not directly assimilated, the treatment of water vapour in the stratosphere is highly variable amongst the reanalyses. For the modern reanalyses, the concentration of water vapour entering the stratosphere is typically controlled by transport and dehydration processes occurring in the forecast model, primarily in the

tropical tropopause layer (TTL). Higher in the stratosphere, chemical production of water vapour through methane oxidation is parameterized in some reanalyses, while others use a simple relaxation of the simulated water vapour field to an observed climatology.

As with water vapour, the treatment of ozone is quite different from reanalysis to reanalysis. The ozone treatment in reanalyses ranges from omitting prognostic ozone and using a climatology in the radiation calculations (NCEP R1/R2), to

using a fully prognostic field with parameterized photochemistry (CFSR, ERA-40, ERA-I, MERRA, MERRA-2), to assimilating ozone with an offline chemical transport model for use in the forecast model radiation calculation (JRA-25, JRA-55).





The primary ozone observations assimilated by reanalyses are satellite nadir UV backscatter-based retrievals of vertically integrated total column ozone (TCO) or broad vertically weighted averages (e.g., SBUV data). These data come from a variety of satellites that have flown since the late 1970's, and reanalyses vary widely in what subset of the available data they assimilate (Figs 1-2). Some further differences exist amongst the reanalyses in their usage of different data versions from the

same satellite instrument, and from different applications of data quality control and filtering. These differences in usage of input data may affect the reanalysis ozone fields.

Additional observation types using spectral ranges outside of the UV (namely microwave and IR) and exploiting different viewing geometries (such as limb-sounding) have been used, particularly by the newest reanalyses (ERA-I, MERRA-2). The assimilation of additional data, particularly vertically resolved data, should improve the quality of the ozone in reanalyses.

However, the assimilation of new data sets could introduce sudden changes in the reanalysis ozone fields, and these transition times should be considered carefully when deriving or analysing long-term trends.

### 2.1 NCEP-NCAR (R1) and NCEP-DOE (R2)

Neither NCEP-NCAR (R1) nor NCEP-DOE (R2) assimilates ozone data (Kalnay et al., 1996; Kanamitsu et al., 2002; Kistler et al., 2001). A climatology of ozone was used for radiation calculations.

Humidity information from satellites is not assimilated in R1 and R2 (Ebisuzaki and Zhang, 2011). In general, the treatment of water vapour is similar in R1 and R2, with a few notable differences. One major difference is that humidity is not output above 300 hPa in R1, whereas it is output up to 10 hPa in R2. Another difference is that only relative humidity is output in R2, whereas in R1 both specific humidity and relative humidity are output. It is worth noting that in R1, specific humidity is a diagnostics variable, computed from relative humidity and temperature. Several fixes and changes were made

in the treatment of clouds in R2, and these result in R2 being ~20% drier than R1 in the tropics at 300 hPa (Kanamitsu et al., 2002). As the focus here is on upper levels, we do not assess humidity fields from R1 or R2. It is worth noting that R1 shows negative long-term humidity trends between 500 and 300 hPa (Paltridge et al., 2009); however, these negative trends appear to reflect suspect radiosonde measurements at these levels and are not found in other reanalyses or satellite data (Dessler and Davis, 2010).

### 25    2.2 CFSR

The Climate Forecast System Reanalysis (CFSR) is a newer NCEP product following the NCEP R1 and R2 reanalyses but with numerous improvements (Saha et al., 2010), including an updated forecast model and data assimilation system. CFSR was originally provided through the end of 2009, but output from the same analysis system was extended through the end of 2010 before transitioning to the CFSv2 analysis system starting in January 2011 (Saha et al., 2014). Because CFSv2 was

intended as a continuation of CFSR, in this paper we refer to both CFSR (i.e., CFSRv1) and CFSv2 as CFSR. However, the system changeover did result in a discontinuity in the water vapour fields that is addressed later in this paper.





CFSR treats ozone as a prognostic variable that is analysed and transported by the forecast model. The CFSR forecast model uses analysed ozone data for radiation calculations. In the forecast model, ozone chemistry is parameterized using production and loss terms generated by the NRL CHEM2D-OPP (McCormack et al., 2006). These production and loss rates are provided as monthly mean zonal means, and are a function of local ozone concentration. The rates do not include the coefficients for temperature and overhead ozone column provided by McCormack et al. (2006), nor heterogeneous chemistry, although late 20[th] century levels of CFCs are used indirectly because CHEM2D-OPP is based on the CHEM2D middle atmospheric photochemical transport model, which includes ODS levels representative of the late twentieth century.

CFSR assimilates version-8 SBUV profile and TCO retrievals (Flynn et al., 2009) from *Nimbus-7* and SBUV/2 profiles and TCO retrievals from *NOAA-9*, *-11*, *-14*, *-16*, *-17*, *-18*, and eventually *NOAA-19* (Figs. 1-2). The ozone layer and TCO values assimilated by CFSR have not been adjusted to account for biases from one satellite to the next, although the use of SBUV version 8 is expected to minimize satellite-to-satellite differences. Despite the fact that CFSR assimilates TCO retrievals and SBUV ozone profiles, differences have been found between CFSR and SBUV(/2) ozone profile data (Saha et al., 2010). Most of these differences are located above 10 hPa, and appear to result from observational background errors that were set too high in the CFSR upper stratosphere by between a factor of 2 (at 10 hPa) and a factor of 60 (at 0.2 hPa). Because of this, assimilated SBUV(/2) ozone layer observations do not alter the CFSR first guess for pressures less than 10 hPa, and the model first guess is used instead. The observational background errors were fixed for CFSv2, starting in 2011.

Water vapour is treated prognostically in CFSR. There are several assimilated observation types that influence the analysis humidity fields in the troposphere, including GNSS-RO bending angles, radiosondes, and satellite radiances. However, as radiosonde humidity data is only assimilated at 250 hPa and greater pressures, there are no specific observations that constrain humidity in the stratosphere. Stratospheric humidity in CFSR is hence primarily governed by physical processes and parameterizations in the model, including dehydration within the TTL. The treatment of water vapour in the model can lead to negative water vapour values around and above the tropopause. These negative values are replaced by small positive values of 0.1 parts per million by volume (ppmv) for the radiation calculations, but are retained in the analysis products. CFSR does not include a parameterization of methane oxidation.

### 2.3 ERA-40

The ERA-40 forecast model included prognostic ozone and a parameterization of photochemical sources and sinks of ozone, as described by Dethof and Hólm (2004). This parameterization of ozone production/loss rates is an updated version of the one proposed by Cariolle and Deque (1986, hereinafter CD86). In CD86, the net ozone production rate is parameterized as a function of the perturbation (relative to climatology) of the local ozone concentration, the local temperature, and the column ozone overhead. Compared to the CD86 formulation, the ozone parameterization in ERA-40 includes an additional term representing heterogeneous chemistry. This loss term scales with the product of the local ozone concentration and the square of the equivalent chlorine concentration, and is only turned on at temperatures below 195 K.





The climatologies and coefficients used in the parameterization are derived from a photochemical model and vary by latitude, pressure, and month. The prescribed chlorine loading varies from year to year, from ~700 parts per trillion (ppt) in 1950 to ~3400 ppt in the 1990s. Instead of the CD86 ozone photochemical equilibrium values, ERA-40 made use of the Fortuin and Langematz (1995) ozone climatology.

5    The prognostic ozone was not used in the radiation calculations, which instead assumed the climatological ozone distribution reported by Fortuin and Langematz (1995). This choice was motivated by concerns that ozone–temperature feedbacks would degrade the temperature analysis if the assimilated ozone observations were of poorer quality than the temperature observations (Dethof and Hólm, 2004).

    ERA-40 assimilated TOMS TCO and SBUV layer ozone retrievals from the end of 1978 onward (Figs. 1-2; See also Table 1, Dethof and Hólm, 2004; Poli, 2010). No ozonesonde measurements were assimilated, and no ozone data at all were assimilated before 1978. Ozone data prior to 1978 are thus primarily products of the photochemical parameterization. In addition, no ozone data were assimilated during 1989 and 1990 because the execution of the first ERA-40 stream (1989–2002) was started before the ozone assimilation scheme was implemented. Ozone background errors were also changed, such that the period January 1991 to October 1996 used different background errors than the rest of ERA-40 (Dethof and Hólm, 2004).

    ERA-40 water vapour products below the diagnosed tropopause are substantially affected by assimilated observations. Three main periods can be identified (Uppala et al., 2005): until 1973, ERA-40 used only conventional in situ surface and radiosonde measurements; from 1973, satellite radiances from VTPR (1973–1978) and the TOVS instruments MSU, SSU, and HIRS (1978–onwards) were used in addition to these conventional data sources; from 1987, 1D-Var retrievals of TCWV from SSM/I radiances were added to the assimilation. Radiosonde humidity measurements were generally used at pressures greater than 300 hPa. No adjustments to the humidity field due to data assimilation were made in ERA-40 above the diagnosed tropopause. Thus, stratospheric water vapour in ERA-40 reflects TTL dehydration, transport, and methane oxidation. The latter was included via a simple stratospheric parameterization, in which WV was gradually relaxed to 6 ppmv at the stratopause (Untch et al., 1998). This relaxation was later found to produce too low WV concentrations at the stratopause as it was based on earlier studies when atmospheric methane levels were lower (Uppala et al., 2005). ERA-40 stratospheric humidity has also been shown to be too low overall, due primarily to a cold bias in TTL temperatures caused by an excessively strong Brewer-Dobson circulation (Oikonomou and O'Neill, 2006).

### 2.4 ERA-Interim

    The treatment of ozone and water vapour in ERA-Interim is very similar to that in ERA-40. Notable differences include additional assimilated datasets and an improved treatment of water vapour in the upper troposphere and lower stratosphere (UTLS). Descriptions of the ozone system and assessments of its quality have been provided by Dee et al. (2011) and Dragani (2011).





As with ERA-40, total ozone from TOMS (Jan 1979–Nov 1989; Jun 1990–Dec 1994; Jun 1996–Dec 2001) and ozone layer averages from SBUV (1979–present) are assimilated (Figs. 1–2). ERA-Interim also assimilates TCO from OMI (Jun 2008– Jan 2009, Mar 2009–present) and SCIAMACHY (Jan 2003–Dec 2008), and ozone profiles from GOME (Jan 1996–Dec 2002), MIPAS (Jan 2003–Mar 2004), and MLS (Jan–Nov 2008, Jun 2009–present). A change in the assimilation of

SBUV ozone profiles was implemented in January 2008. Before January 2008, assimilated SBUV profiles were low vertical resolution products derived over six vertical layers (0.1–1 hPa, 1–2 hPa, 2–4 hPa, 4–8 hPa, 8–16 hPa and 16 hPa–surface) from NOAA version 6 (v6) retrievals. These data were replaced by native 21-vertical-level SBUV profiles from v8 retrievals. The assimilation of ozone profile retrievals from Aura MLS started in 2008 (Fig. 2) using the reprocessed v2.2 MLS retrievals and carried on with the near-real-time v3 product from June 2009 onwards.

The ozone forecast model used in ERA-Interim has the same basic formulation as that used in ERA-40 but some aspects of the parameterization have been upgraded substantially, especially the regression coefficients. An account of the changes is provided by Cariolle and Teyssédre (2007). As in ERA-40, the radiation scheme in ERA-Interim does not use the prognostic ozone field.

A preliminary assessment of the temperature and wind fields revealed unrealistic temperature and horizontal wind

increments generated near the stratopause by the 4D-Var assimilation scheme in an attempt to accommodate large local adjustments in ozone concentrations (Dee, 2008; Dragani, 2011). As an ozone bias correction was not available in ERA-Interim to limit the detrimental effect of ozone assimilation on temperature and wind fields, the sensitivity of the latter to ozone changes was switched off in ERA-Interim. This change affected the period from 1 February 1996 onwards and the ten years from 1979 through 1988 that were run at a later stage.

Through December 1995, ERA-Interim ozone analyses perform better than their ERA-40 counterparts with respect to independent ozone observations in the upper troposphere and lower stratosphere, but perform slightly worse on average in the middle stratosphere (Dee et al., 2011). The assimilation of GOME ozone profiles (Jan 1996–Dec 2002) improves the agreement between ERA-Interim analyses and independent data, such that ERA-Interim outperforms ERA-40 throughout the atmosphere (including the middle stratosphere) from January 1996 through the end of ERA-40 in September 2002 (Dragani,

25     2011).

The ERA-Interim humidity analysis is substantially modified from that in ERA-40 due to changes in both model physics and assimilated observations. A non-linear transformation of the humidity control variable was introduced to make humidity background errors more Gaussian (Uppala et al., 2005; Hólm, 2003; Hólm et al., 2002). This transformation normalizes relative humidity increments by a factor that depends on background estimates of relative humidity and vertical level. A 1D-

Var assimilation of rain-affected radiances over oceans was also added as part of the 4D-Var outer loop (Dee et al., 2011), which helps to constrain the spatial distribution of total column water vapour (TCWV). The ERA-Interim humidity analysis also benefits from several changes in the model physics, including changes in the convection scheme that lead to increased convective precipitation (particularly at night), reduced tropical wind errors, and a better representation of the diurnal phasing of precipitation events (Bechtold et al., 2004). The non-convective cloud scheme was also been updated.



Perhaps of most relevance for humidity in the UTLS, the revised cloud scheme contains a new parameterization that allows supersaturation with respect to ice in the cloud-free portions of grid cells with temperatures less than 250 K (Tompkins et al., 2007). The inclusion of this parameterization results in substantial increases in relative humidity in the upper troposphere and in the stratospheric polar cap relative to ERA-40 (Dee et al., 2011). Methane oxidation in the

stratosphere is included via a parameterization like the one used in ERA-40 but with relaxation to 6.8 ppmv at the stratopause (rather than 6 ppmv as in ERA-40), based on an analysis of UARS data by Randel et al. (1998).

As with ERA-40, no adjustments due to data assimilation are applied in the stratosphere (above the diagnosed tropopause). ERA-interim tropospheric humidity is affected by the assimilation of radiosonde humidity measurements, radiances from the TOVS (through 5 Sep 2006) and ATOVS (from Aug 1998) instrument suites, and TCWV retrievals based on rain-affected

radiances from SSM/I (from Aug 1987). Recent ERA-Interim humidity analyses may also be affected by the assimilation of GNSS-RO bending angles (from May 2001) and/or AIRS all-sky radiances (from April 2004).

### 2.5 JRA-25 and JRA-55

Ozone observations were not assimilated directly in the JRA-25 and JRA-55 systems (Kobayashi et al., 2015; Onogi et al., 2007). Instead, daily three-dimensional ozone fields were produced separately and provided to the JRA forecast model (i.e.

to the radiation scheme). Daily ozone fields in JRA-55 for 1978 and earlier are interpolated in time from a monthly mean climatology for 1980–1984. Daily ozone fields in both systems for 1979 and later are produced using an offline chemistry climate model (MRI-CCM1, Shibata et al., 2005) that assimilated satellite observations of TCO using a nudging scheme. Assimilated TCO retrievals are taken from TOMS on Nimbus-7 and other satellites for the period 1979–2004 and from Aura OMI after the beginning of 2005. Different versions of MRI-CCM1 and different preparations of the ozone fields have been

used for JRA-25 and JRA-55. For JRA-25, MRI-CCM1 output were also nudged to climatological ozone vertical profiles to account for a known bias in tropospheric ozone that produces a bias in stratospheric ozone after nudging to observations of total ozone. This procedure produced reasonable peak ozone-layer values in the final ozone product. This vertical-profile nudging was not necessary for JRA-55, which used an updated version of MRI-CCM1. JRA-55 produces improved peak values in vertical ozone profiles relative to JRA-25, as well as a clear ozone qausi-biennial oscillation (QBO) signature.

As with other modern reanalyses, JRA-25 and JRA-55 humidity fields are affected by the assimilation of radiosonde humidity measurements and satellite radiances. The JRA-25 assimilation analysed the logarithm of specific humidity (Onogi et al., 2007). Stratospheric humidity was dry-biased and generally decreased with time in JRA-25, in part due to the lack of parameterized methane oxidation. The JRA-25 forecast model radiation calculations assumed a constant value of 2.5 ppmv in the stratosphere. Water vapour in the UTLS shows evidence of discontinuities at the start of 1991, which corresponds to

the transition between the two major processing streams of JRA-25. Onogi et al. (2007) reported sudden jumps of +0.7 ppmv at 150 hPa and +0.9 ppmv at 100 hPa associated with this transition.

The treatment of water vapor in JRA-55 is similar in most respects to that in JRA-25. JRA-55 does not contain a parameterization of methane oxidation. Differences include a change in the upper boundary above which the vertical





correlations of humidity background errors are set to zero, preventing spurious analysis increments at higher levels. This boundary is set at 5 hPa in JRA-55, relative to 50 hPa in JRA-25. Forecast model radiation calculations in JRA-55 use an annual mean climatology of stratospheric water vapour derived from UARS HALOE and UARS MLS measurements made during 1991–1997 in the stratosphere, rather than the constant 2.5 ppmv used in JRA-25. The introduction of an improved

radiation scheme in JRA-55 greatly reduced lower stratospheric negative temperature biases that were present in JRA-25 during the TOVS period before 1998 (Kobayashi et al., 2015; Fujiwara et al., 2017), which may have beneficial impacts on JRA-55 stratospheric humidity products. Water vapour concentrations at pressures less than 100 hPa are not provided in the standard pressure-level products of these two reanalyses (although these concentrations are provided in model-level products), and are therefore not evaluated in this paper.

**2.6 MERRA**

Ozone is a prognostic variable in MERRA, and is subjected to assimilation, transport by assimilated winds (more precisely, the odd-oxygen family is the transported species), and parameterized chemistry. The MERRA general circulation model (GCM) uses a simple chemistry scheme that applies monthly zonal mean ozone production and loss rates derived from a 2-dimensional chemistry model (Stajner et al., 2008). Ozone data assimilated in the reanalysis include partial columns

and total ozone (defined as the sum of layer values in a profile) from a series of SBUV instruments (Flynn et al., 2009) on various NOAA platforms (Figs. 1–2). Version 8 of the SBUV retrievals (Flynn, 2007) is used but the native 21 vertical layers are combined into 12 layers (each 5 km deep) prior to assimilation. All other assimilated data, including radiance observations, are explicitly prevented from impacting the ozone analysis directly. Since SBUV sensors measure backscatter solar ultraviolet radiation, only daytime observations are available; wintertime ozone in polar regions is thus poorly

constrained by observations. Early NOAA satellites experienced orbital drifts that resulted in reduced daylight coverage over time. For example, the equatorial crossing time for NOAA-11 drifted from ~2PM in 1989 to ~5PM five years later, leading to limited SBUV coverage in 1994 (ozone observations were entirely unavailable south of 30°S during that austral winter). A similar orbital drift in the NOAA-17 satellite impacted the quality of the MERRA ozone products in 2012 before the introduction of observations from NOAA-19 SBUV in 2013. Outside of the exceptions described above and occasional short

temporal gaps, SBUV provides good coverage of the sunlit atmosphere.

Background error standard deviations for ozone are specified as ~4% of the global mean ozone on a given model level. Horizontal background error correlation lengths vary from ~400 km in the troposphere to ~800 km at the model top. Assimilated ozone fields are fed into the forecast model radiation scheme and are used in the radiative transfer model for radiance assimilation.

Water vapour is also a prognostic assimilated variable in MERRA; however, unlike ozone, moisture fields in the stratosphere are relaxed to a 2-D monthly climatology with a relaxation time of 3 days. This climatology is derived from water vapour observations made by the UARS HALOE and Aura MLS instruments (e.g., Rienecker et al., 2011 and references therein). This climatological constraint is introduced gradually over the layer between the model tropopause and



hPa, where pressure-dependent blending between the climatology and the GCM water vapour is applied. Water vapour above the tropopause does not undergo physically meaningful variations on timescales longer than the 3-day relaxation timescale except in the lowermost stratosphere where the climatology is given a smaller weight. No attempt was made to account for methane oxidation or trends in stratospheric methane concentrations.

MERRA assimilates specific humidity measurements from radiosondes at pressures above 300 hPa and marine surface observations. Moisture fields are affected by microwave radiance data from SSM/I and AMSU-B/MHS, infrared radiances from HIRS, the GOES Sounder, and AIRS, and rain rates derived from TMI and SSM/I. Background error statistics for water vapour were derived using the National Meteorological Center method and applied using a recursive filters methodology (Wu et al., 2002). The moisture control variable is pseudo-relative humidity (Dee and Da Silva, 2003).

**2.7 MERRA-2**

The key differences between the treatment of ozone in MERRA-2 and that in MERRA are in the observing system and background error covariances. From January 1980 to September 2004, MERRA-2 assimilates v8.6 SBUV retrievals of partial columns on a 21-layer vertical grid (Bhartia et al., 2013) and total ozone computed as the sum of individual layer values. Compared to the v8 retrievals used in MERRA, the v8.6 algorithm uses upgraded ozone cross-sections and an
improved cloud height climatology. These updates result in better agreement with independent ozone data and make SBUV more suitable for long-term climatologies (Frith et al., 2014; McPeters et al., 2013). Starting in October 2004, SBUV data was replaced by a combination of TCO from Aura OMI (Levelt et al., 2006) and stratospheric profiles from Aura MLS (Waters et al., 2006). The OMI data consist of TCO retrievals from collection 3 and are based on the v8.5 retrieval algorithm, which is an improvement of the v8.0 algorithm extensively evaluated by McPeters et al. (2008). The assimilation
algorithm makes use of the OMI averaging kernels to account for the sensitivity of these measurements to clouds in the lower troposphere (Wargan et al., 2015). MLS data are from v2.2 between October 2004 and May 2015 and v4.2 (Livesey et al., 2017) afterwards. Users of the MERRA-2 ozone product should therefore be aware that the reanalysis record may show a discontinuity in 2004 with two distinct periods as follows: the SBUV period (1980–September 2004) and the EOS Aura period (from October 2004 onward). The analysis is expected to be of higher quality during the latter period due to the
higher vertical resolution of Aura MLS profiles relative to SBUV profiles and the availability of MLS observations during night.

Ozone background error variance in the MERRA-2 model follows Wargan et al. (2015). The background error standard deviation at each grid point is proportional to the background ozone at that point and time. This approach introduces a flow dependence into the assumed background errors and allows a more accurate representation of shallow structures in the ozone
fields, especially in the UTLS. As in MERRA, the ozone analyses are radiatively active tracers in both the forecast model and the radiative transfer model used for assimilation of satellite radiances. Bosilovich et al. (2015) provided a preliminary evaluation of the MERRA-2 ozone product. A more comprehensive description and validation, including comparisons with MERRA, is given in Wargan et al. (2017)





The treatment of stratospheric water vapour in MERRA-2 is similar to that in MERRA, with a 3-day relaxation to the same climatological annual cycle. The main innovation is the introduction of additional global constraints that ensure the conservation of the dry mass of the atmosphere and rescale the water vapour tendency to remove the globally integrated mean from the analysis increment (Takacs et al., 2016).

In addition to the moisture data assimilated in MERRA, MERRA-2 assimilates GNSS-RO data and radiances from the recently introduced infrared sensors IASI, CrIS, and SEVIRI. Radiances from these recent IR instruments are not highly sensitive to stratospheric water vapour. Stratospheric water vapour is therefore not intentionally adjusted by the assimilation of these observations but may be affected in small ways. Changes in the MERRA-2 observing system relative to MERRA are described in more detail by Bosilovich et al. (2015) and McCarty et al. (2016). The moisture control variable in the

MERRA-2 assimilation scheme is pseudo-relative humidity normalized by the background error standard deviation. Background error covariances used in MERRA-2 have been significantly retuned relative to those used in MERRA (Bosilovich et al., 2015).

## 3 Data

In this section, we describe the approach we use to process the reanalysis ozone and water vapour fields, and the

observations used to evaluate them. We note that some of these observational data are assimilated by the reanalyses. While comparisons between reanalyses and observations would ideally be based on independent observations, this is not always possible given the paucity of water vapour and ozone data in parts of the atmosphere. However, comparison to assimilated observations can serve a useful purpose by providing an internal consistency check on the ability of reanalysis data assimilation systems to exploit the data they assimilate.

### 3.1 Reanalysis data processing

Most of the comparisons presented in this paper are based on monthly mean reanalysis fields calculated from the "pressure level" data sets provided by each reanalysis centre, and processed into a standardized format as part of the CREATE project (https://esgf.nccs.nasa.gov/projects/create-ip/). To facilitate intercomparison amongst reanalyses, the pressure level-based datasets have been re-gridded to a common horizontal grid ($2.5°$ lon x $2.5°$ lat) and a common set of 26 pressure levels

(1000, 925, 850, 700, 600, 500, 400, 300, 250, 200, 150, 100, 70, 50, 30, 20, 10, 7, 5, 3, 2, 1, .7, .5, .3, .1 hPa). Unless otherwise noted, climatological comparisons follow the WMO convention in using the 30-year 1981–2010 climatological norm (Arguez and Vose, 2011).

Reanalysis TCO data are monthly means computed from the 6-hourly model level data. All of the models provided 6-hourly TCO on various native horizontal grids, except for JRA-25. For JRA-25, 6-hourly ozone mass mixing ratios were

provided on model levels. The mixing ratios were integrated for each horizontal grid point to get TCO, and then monthly means were computed. For each reanalysis, the climatologies and departures from climatology were calculated and are





presented on each data set's native horizontal grid. For comparisons to the SBUV and TOMS/OMI data, each model was interpolated to the native horizontal grid of each of the observational data sets.

**3.2 SBUV and TOMS/OMI total column ozone**

Two datasets are used to evaluate the total column ozone in the reanalyses. The first is the SBUV Merged Ozone Data Set
(Frith et al., 2014). The second is a combination of TOMS and Aura OMI OMTO3d total ozone observations (Bhartia and Wellemeyer, 2002). These two data sets provide a long, coherent span of observations for evaluation. TOMS data were processed using the TOMS V8 algorithm, while the OMI and SBUV data were processed using the TOMS V8.6 algorithm. Because data from SBUV and TOMS (and in many cases OMI) are assimilated by most of the reanalyses, these comparisons are not independent.

**3.3 SPARC Data Initiative limb satellite observations**

The SPARC Data Initiative (Fueglistaler et al., 2009; Gettelman et al., 2011) offers monthly mean zonal mean climatologies of ozone (Neu et al., 2014; Tegtmeier et al., 2013) and water vapour (Hegglin et al., 2013) from an international suite of satellite limb sounders. The zonal monthly mean climatologies have undergone a comprehensive quality assessment and are suitable for climatological comparisons of the vertical distribution and interannual variability of
these constituents in reanalyses on monthly to multi-annual timescales. We use a subset of the instrumental records available, as specified below.

The observational multi-instrument mean (MIM) for ozone averaged over 2005–2010 is derived using the SPARC Data Initiative (in the following abbreviated as SDI) zonal monthly mean climatologies from ACE-FTS, Aura MLS, MIPAS, and OSIRIS. These instruments provide data for the full 6 years considered and show inter-instrument differences with respect to
the MIM that are generally smaller than ±5% throughout most of the stratosphere. Hence, temporal inhomogeneities that could affect the MIM are avoided and the standard deviation in the MIM is relatively small. Differences from the MIM in the lower mesosphere and tropical lower stratosphere are somewhat higher (±10%) (Tegtmeier et al., 2013). The evaluation of the ozone QBO signal for 2005–2010 is based on the instruments OSIRIS, GOMOS, and Aura MLS, which produce the most consistent QBO signals (Tegtmeier et al., 2013).

The observational MIM for water vapour averaged over 2005–2010 is derived using the SDI zonal monthly mean climatologies from Aura MLS, MIPAS, ACE-FTS, and SCIAMACHY. These instruments show inter-instrument differences that are generally within ±5% of the MIM throughout most of the stratosphere (Hegglin et al., 2013). Differences from the MIM in the tropical upper troposphere increase to ±20%.

**3.4 Aura MLS satellite data**

The evolution of ozone in the reanalyses is compared with that observed by Aura MLS. This instrument measures millimeter- and submillimeter-wavelength thermal emission from Earth's atmosphere using a limb viewing geometry. Waters





et al. (2006) provide detailed information on the measurement technique and the Aura MLS instrument. Vertical profiles are measured every 165 km along the suborbital track with an along-track horizontal resolution of 200~500 km and a cross-track footprint of 3~9 km. Here we use version 4.2 (hereafter v4) MLS ozone measurements from September 2004 through December 2013. The quality of the MLS v4 data has been described by Livesey et al. (2017). The vertical resolution of MLS

ozone is about 3 km and the single-profile precision varies with height from approximately 0.03 ppmv at 100 hPa to 0.2 ppmv at 1 hPa. The v4 MLS data are quality-screened as recommended by Livesey et al. (2017). V4 stratospheric (pressures less than 100 hPa) ozone values are within ~2% of those in version 2.2 (v2), which is the version assimilated in MERRA-2 (until 31 May 2015, after which v4 data are used) and ERA-Interim. At pressures greater than 100 hPa, v4 MLS ozone shows high and low biases with respect to v2 at alternating levels, indicating improvement of vertical oscillations seen in v2

(Livesey et al., 2017) and v3 (Yan et al., 2016).

### 3.5 SWOOSH merged limb satellite data record

The Stratospheric Water and Ozone Satellite Homogenized (SWOOSH) database is a monthly-mean record of vertically resolved ozone and water vapour data from a subset of limb profiling satellite instruments operating since the 1980s (Davis et al., 2016).  SWOOSH includes individual satellite source data from SAGE-II (v7), SAGE-III (v4), UARS MLS (v5/6),

UARS HALOE (v19), and Aura MLS (v4.2), as well as a merged data product. A key aspect of the merged product is that the source records are homogenized to account for inter-satellite biases and to minimize artificial jumps in the record. The homogenization process involves adjusting the satellite data records to a "reference" satellite using coincident observations during time periods of instrument overlap. SWOOSH uses SAGE-II as the reference for ozone and Aura MLS as the reference for water vapour. SWOOSH merged product data are used for timeseries evaluations that start before 2004, prior to

the availability of Aura MLS. After August 2004, the SWOOSH merged product is essentially the same as the v4.2 Aura MLS data.

## 4. Evaluation of reanalysis ozone products

### 4.1 Total column ozone seasonal cycle

In this section, we compare SBUV TCO data to reanalysis products over the 1981–2010 climatology period. Figure 3

shows the seasonal cycle in total column ozone from SBUV as a function of latitude and month. Also shown are the differences between TOMS/OMI and SBUV, and between the different reanalyses and SBUV. The climatological TCO fields of the TOMS/OMI and the reanalyses are given as line contours in the difference plots. Supplementary Figure S1 shows the equivalent comparison for TOMS/OMI data. The reanalyses all reproduce the major features of the seasonal cycle and latitudinal distribution of TCO. This agreement is not surprising given that all of reanalyses shown in Fig. 3 assimilate

TCO data from one of the two satellites (Fig. 1). As such, the comparisons here do not represent independent validation of ozone in reanalyses but rather represent a test of the internal consistency of the ozone data assimilation system. Hence it is





not surprising that MERRA and MERRA-2 generally perform better against SBUV than against TOMS/OMI, while ERA-Interim and JRA-55 generally perform better against TOMS/OMI than against SBUV, since MERRA and MERRA-2 assimilate SBUV (but not TOMS/OMI), while ERA-Interim and JRA-55 primarily assimilate TOMS/OMI (but not SBUV).

Although the reanalysis TCO fields look quite similar, a handful of widespread biases are revealed by considering the differences between reanalyses and observations. The agreement between the two observational TCO datasets is within approximately ±6 DU (2~3%), with SBUV generally having smaller values in the tropics and larger values at high latitudes relative to TOMS/OMI. Differences between the reanalyses and the TCO observations are generally slightly larger than the difference between the two observational datasets. ERA-40 produces substantially larger TCO values than observed, particularly at higher latitudes. JRA-25 contains significantly smaller TCO values than observed (~10 DU less), except during the springtime at high southern latitudes.

For reanalyses that only (or mainly) assimilate UV-based retrievals, the winter hemisphere high latitudes remain largely unconstrained by data assimilation. The impact of the TCO observations may also be limited by filtering choices. For example, assimilated observations are filtered to exclude low solar elevation angles (less than 10° for TOMS and less than 6° for SBUV) in both ERA-40 and ERA-Interim. This filtering further limits observational impacts on the ozone analyses at higher latitudes. Hence, for ERA-Interim, before the start of the Aura MLS assimilation in 2008, high latitude ozone fields essentially reflect the effects of transport and the ozone parameterization used. For ERA-40, Dethof and Hólm (2004) showed that the ozone model produces high biases in ozone concentrations at high latitudes ranging from ~20 DU in the summer hemisphere to ~50 DU in the winter hemisphere, which is broadly consistent with the comparison shown in Fig. 3.

### 4.2 Zonal mean ozone cross-sections

In this section, we compare zonal mean multi-annual mean cross sections of ozone between the different reanalyses and the SDI MIM. We perform the comparison for 2005–2010 using the subset of instruments described in Sect. 3.3. This shorter period has been chosen to avoid sampling issues that could be introduced by changes in instrument availability, which could alter sampling patterns, or trends in the constituents, such as the increase in ozone depletion from the 1970s to the mid 1990s. ERA-40 is excluded from this and all other comparisons with the SDI MIM because it ended in 2002.

Figure 4 shows multi-annual zonal mean ozone from the SDI MIM and the relative differences between each reanalysis and the SDI MIM (calculated as $100*(R_i - MIM)/MIM$, where $R_i$ is the reanalysis field). Also indicated using contours are the climatological ozone distributions of the reanalyses. The reanalyses all capture the general zonal mean distribution of ozone, including the global maximum in ozone volume mixing ratio in the tropical middle stratosphere and the tropopause-following isopleths immediately above the tropopause. Among the reanalyses, MERRA-2 best reproduces this overall structure, with relative differences within ±5% throughout the middle and upper stratosphere. MERRA, CFSR, and ERA-Interim also perform generally well, but with MERRA overestimating concentrations in the ozone maximum (~10 hPa) relative to the SDI MIM. ERA-Interim shows relatively good agreement in the middle stratosphere with biases smaller than ±5% but includes a low bias with magnitudes greater than 10% in the upper stratosphere. All reanalyses show biases



exceeding ±10% in the lowermost stratosphere, at pressures greater than 100 hPa. JRA-55 shows evident improvement relative to JRA-25. The latter included a strong positive bias (greater than 20% relative to the MIM) in the lower and middle stratosphere and a strong negative bias (>20%) in the upper stratosphere. JRA-55 still shows a negative bias in the upper stratosphere, but the magnitude is approximately halved relative to that in JRA-25. It is worth noting that the diurnal cycle in

ozone has not been explicitly accounted for in the observational MIM. Neglecting the diurnal cycle potentially contributes to differences between the reanalyses and observations in the upper stratosphere and lower mesosphere.

All reanalyses (except JRA-55) produce a positive bias in ozone in the Southern Hemisphere (SH) lower stratosphere. This indicates an inability to simulate Antarctic ozone depletion accurately due to a combined effect of limited data coverage, data filtering, and limitations of the reanalyses' chemistry schemes at high latitudes (Sect. 4.1). A dipole is apparent in the

CSFR and ERA-Interim biases, with a high bias near ~100 hPa located below a low bias near ~10 hPa. This dipole may reflect a lack of information about the vertical location of the ozone hole in the TCO and SBUV observations assimilated by these systems. In contrast, MERRA includes a significant high bias (>10%) at Southern high latitudes that extends throughout the stratosphere.

### 4.3 Ozone monthly mean vertical profiles and seasonal cycles

Figures 5a and b show vertical profiles of ozone for January (2005–2010 average) for the reanalyses and the SDI MIM at two different latitudes, 40°N and 70°S, respectively, along with the relative differences for each reanalysis with respect to the MIM. In addition, Figures 5c and d show the seasonal cycles of ozone for three different pressure levels at 40°N and 70°S, respectively. The vertical profiles and the seasonal cycles reveal seasonal information on reanalyses-observation differences that expands upon the annual zonal mean evaluation presented in Sect. 4.2. In general, the results shown

reinforce the conclusions of the previous section.

Most reanalyses resolve the vertical distribution in January reasonably well at both latitudes, in particular in the middle stratosphere between around 50 and 5 hPa. MERRA-2, MERRA, and CFSR perform particularly well. JRA-25, on the other hand, is a clear outlier that places the ozone maximum too low in the profile and produces too little ozone above the maximum. JRA-55 and ERA-Interim also underestimate ozone concentrations in the upper stratosphere by between 10 and

20% but are not as strongly biased as JRA-25 (which produces differences of more than 20%). All reanalyses show larger percentage differences from the MIM in the lower part of the profile at pressures greater than 100 hPa. The reanalyses seem to overestimate ozone at around 150 hPa by 20% in the Southern high latitudes, possibly related to not capturing accurately enough the extent of ozone depletion during spring. Below 200 hPa at both latitudes, all reanalyses underestimate observed ozone values.

The agreement between the reanalyses and observations varies by month, as can be seen in Figures 5c and d, which show the annual cycle for selected pressure levels (150, 50, and 10 hPa) and somewhat extended latitude bands of 30°N-50°N and 60°S-80°S, respectively. The agreement in the ozone seasonal cycle between the SDI observations and the reanalyses is better in the Northern Hemisphere (NH) mid-latitudes (where the seasonal cycles have a simple sinusoidal structure) than in





the SH high latitudes. In the NH at 50 and 150 hPa, ozone reaches its annual maximum during boreal spring and its annual minimum during autumn, attributable to the strong seasonality in the Brewer-Dobson circulation. The seasonal cycle is shifted at 10 hPa, with a maximum in summer and a minimum in winter, attributable mostly to ozone photochemistry. Most of the reanalyses produce a fairly accurate ozone evolution at these levels with exceptions as follows: At 150 hPa, JRA-55

shows a strong low bias when compared to both observations and the other reanalyses during the NH winter/spring months. All the other reanalyses tend to overestimate the absolute ozone values, but agree rather well with the seasonal cycle in the observations in terms of amplitude and phase. At 50 hPa, the seasonal cycle produced by JRA-55 shows a more gradual decline in ozone concentrations into autumn relative to both observations and other reanalyses, and JRA-25 shows too high ozone mean values throughout the year. At 10 hPa, JRA-25 produces a seasonal cycle that has a bi-modal structure more

consistent with that observed in the tropics (not shown). ERA-Interim, MERRA, and CFSR at 10 hPa tend to overestimate ozone during spring and early summer, while JRA-55 tends to underestimate ozone during fall and winter.

Seasonal cycles in SH high latitudes have a more complex structure than those in the NH mid-latitudes due to generally weaker downwelling in the Brewer–Dobson circulation and the influence of Antarctic ozone depletion. As a consequence, the reanalyses have more difficulty in capturing the seasonal cycle. At 10 hPa, MERRA-2 shows the best agreement with the

observations. CFSR also follows the observations relatively well, but overestimates the amplitude of the seasonal cycle, primarily because of values that are too low during May through July. MERRA and JRA-25 are outliers in that they do not contain the strong annual minimum observed during late boreal autumn and early winter. At 50 hPa, MERRA and JRA-25 agree better with observations than at 10 hPa, but still underestimate austral springtime ozone depletion. Finally, at 150 hPa, the seasonality in the reanalyses varies widely and is inconsistent with that in the observations, with the exception of

MERRA, which produces the most realistic seasonal cycle amplitude. MERRA-2 shows the closest agreement with observations at all levels except for 150 hPa, which is the next to lowest valid level of the MLS v2.2 ozone retrievals that it assimilates.

**4.4 Ozone interannual variability**

Figure 6 shows time series of interannual variability of ozone and its anomalies in the SDI MIM and reanalyses during

2005–2010. The anomalies, which are calculated for each reanalysis by subtracting multi-year monthly means averaged over 2005–2010 from the monthly mean timeseries, are a good indicator of how well physical processes (such as transport) are represented in reanalyses. Time series are shown for the SH high latitudes (averaged over 60°S-80°S) at 50 hPa, and for the NH mid-latitudes (40°N-60°N) at 150 and 10 hPa. In all cases, MERRA-2 produces the closest match with the SDI MIM in terms of both the absolute values and the structure of its interannual variability. This agreement highlights the benefit of

assimilating vertical profile observations from a limb-viewing satellite instrument. Although it has to be noted that the comparison is not done against truly independent observations in this case, since Aura MLS is included in the SDI MIM. MERRA-2 is an evident improvement over MERRA, which tends to disagree with the absolute ozone values of the observations at 150 hPa and to overestimate them at 10 hPa, and to underestimate interannual variability at both levels in the



NH mid-latitudes. JRA-55 also shows clear improvement relative to JRA-25 with respect to the amplitude and structure of interannual variability, at least at 10 hPa in the NH mid-latitudes. Large excursions seen in JRA-25, such as the sudden drop in ozone at the beginning of 2008, are not present in JRA-55 or in the observations.

Although ERA-Interim ozone mean values mostly agree well with observations, the amplitude of its interannual variability is larger than observed. In particular, ERA-Interim overestimates the negative anomaly in NH midlatitudes at 10 hPa, and the positive anomaly in SH high latitudes at 50 hPa during 2008. The largest differences appear to affect ERA-Interim from mid-2009 when the assimilation of Aura MLS data restarted with the (v3) NRT product after months of data unavailability. CSFR also produces large interannual excursions during certain years (e.g., during spring 2006 and 2007 at 50 hPa in SH high latitudes). This issue may be related to SBUV only offering measurements between September to March, so that the assimilation system is not well constrained during the remainder of the year.

### 4.5 Ozone time series in equivalent latitude coordinates

Equivalent latitude (EqL) is a common vortex-centred coordinate used in studies of the stratosphere (e.g., Butchart and Remsberg, 1986; Manney et al., 1999; and references therein). This coordinate is also useful as a geophysically-based coordinate in the UTLS (e.g., Santee et al., 2011), although interpretation becomes more complicated in this context (e.g., Manney et al., 2011; Pan et al., 2012). The equivalent latitude of a potential vorticity (PV) contour is defined as the latitude of a circle centred about the pole enclosing the same area as the PV contour (see Hegglin et al., 2006 for a visual illustration). Figure 7 shows the time series of v4 MLS ozone (Sect. 3.4) for late 2004 through 2013 in the lower stratosphere (520 K), along with differences between MERRA, MERRA-2, ERA-Interim, CFSR, and JRA-55 and MLS ozone at the same level. MLS ozone is interpolated to isentropic surfaces using temperatures from MERRA. The EqL ozone time series are then produced using a weighted average of MLS data in EqL and time, with data also weighted by measurement precision (e.g., Manney et al., 2007; Manney et al., 1999). Figures S2-S3 in the Supplement show the equivalent evaluation for the 350K and 850 K potential temperature levels.

Figure 7 reveals that MERRA-2 matches MLS more closely over the full period than do the other reanalyses. This is expected because the stratospheric ozone reanalyses in MERRA-2 are largely constrained by the MLS stratospheric ozone profiles (v2 for the period shown here). This agreement is especially apparent during Antarctic winter and spring, when other assimilated ozone products (e.g., SBUV/2 and TOMS) cannot provide measurements due to darkness and simplified chemical parameterizations cannot adequately represent heterogeneous loss processes. The improved vertical resolution of MLS relative to SBUV/2 also better constrains the structure of the ozone hole, which is vertically limited. ERA-Interim also shows close agreement with MLS during the periods when it assimilates MLS ozone products (2008 and mid-2009 through present). Biases in the reanalyses that do not assimilate MLS and OMI ozone vary in magnitude and sign, not only among the reanalyses but also with altitude and latitude (see also Figs. S2-S3). High biases in MERRA and CFSR ozone during Arctic winter may be partially related to inadequate representations of ozone chemistry and an overall lack of measurements. We speculate that the latter is dominant due to the appearance of these biases even during years with minimal observed



chemical ozone loss. JRA-55 biases increase strongly with altitude (cf., Figs. S2-S3), becoming even larger in the upper stratosphere. These large biases suggest that column ozone alone is insufficient to properly constrain the vertical distribution of the ozone analyses.

**4.6 Ozone quasi-biennial oscillation signals**

Variations in transport and chemistry associated with the quasi-biennial oscillation (QBO) in tropical zonal wind are among the largest influences on interannual variability in equatorial ozone. The QBO signal in tropical ozone has a double-peaked structure with maxima in the lower (50–20 hPa) and the middle-to-upper (10–2 hPa) stratosphere (Hasebe, 1994; Zawodny and Mccormick, 1991). Ozone is mainly under dynamical control below 15 hPa, where the QBO signal results primarily from changes in ozone transport due to the QBO-induced residual circulation. In contrast, ozone is under

photochemical control above 15 hPa. The QBO signal in these upper levels is understood to arise from a combination of QBO-induced temperature variations (Ling and London, 1986; Zawodny and Mccormick, 1991) and QBO-induced variability in the transport of $NO_y$ (Chipperfield et al., 1994). A realistic characterization of the time–altitude QBO structure is an important aspect of physical consistency in ozone data sets.

   Figure 8 shows time–altitude cross-sections of deseasonalized ozone anomalies from 2005 to 2010 from the SDI MIM,

along with the differences between the ozone anomaly fields from the reanalyses and the SDI MIM. The climatological QBO anomaly fields of the reanalyses are given as contours in the difference plots. Combined ozone measurements from the limb-viewing satellite instruments show a downward propagating QBO ozone signal with a shift in the phase around 15 hPa. All reanalyses exhibit some degree of quasi-biennial variability; however, differences are evident in the phase, amplitude, vertical extent, and downward propagation of these signals. The largest deviations from observations are in JRA-25, which

displays positive anomalies from 2005 to mid-2007 followed by negative anomalies from mid-2007 through 2010 in place of the QBO signal above 15 hPa. In contrast, ERA-Interim shows predominantly negative anomalies in the 100–10 hPa pressure range before 2008 and positive anomalies afterwards. The changes in ERA-Interim coincide with the beginning of the assimilation of Aura MLS profiles beginning in 2008, which caused a shift to positive anomalies. Negative anomalies are present during the first half of 2009 when no MLS data were assimilated, followed by positive anomalies after the

reintroduction of MLS data in June 2009 (Sect. 2.5). CFSR and MERRA produce anomalies that are roughly consistent in amplitude and frequency with the QBO ozone signal in the satellite data. However, no clear downward propagation is apparent in this signal despite the existence of a downward-propagating signal in SBUV data (McLinden et al., 2009), the only vertically-resolved ozone measurements assimilated by CFSR and MERRA. The vertical structure of the anomalies is also shifted. Instead of a pair of peaks in the lower stratosphere (50–20 hPa) and middle-to-upper stratosphere (10–2 hPa), a

single peak emerges near 15 hPa. JRA-55 and MERRA-2 produce a phase and amplitude of QBO variability like those observed in the satellite data. Overall, the features of the QBO (including the downward propagation) are much improved in MERRA-2 relative to MERRA (Coy et al., 2016), and in JRA-55 relative to JRA-25. Nearly all reanalysis data sets extend



the QBO ozone signal to altitudes below 100 hPa; this upper tropospheric signal is not present (or not captured) in the satellite observations.

### 4.7 Ozone hole area

The Antarctic "ozone hole" is a region of severe ozone depletion that starts in late August or early September and lasts
until November or early December. The ozone hole is commonly defined as the area within the 220 DU TCO contour. Figure 9 shows average ozone hole areas based on TOMS/OMI observations and six reanalyses during 1981–2010. The average is computed over 21 September–20 October of each year. This period is chosen to avoid the partial coverage of the SH high latitudes that occurs in TOMS/OMI data during the early part of September. Observationally based ozone hole areas are larger than those produced by the reanalyses in almost all years between 1981 and 2002. The systematic negative bias in
reanalysis-based ozone hole areas is consistent with reanalyses generally underestimating ozone loss. Most of the models track the observations well starting in 2003. This is not a truly independent comparison (all reanalyses except for MERRA assimilate TOMS and/or OMI observations); however, it does show the general consistency among most reanalyses in reproducing realistic interannual and decadal changes in the size of the Antarctic ozone hole, except for a few outliers discussed below.

The newer reanalyses (MERRA-2, ERA-Interim, JRA-55, and CSFR) are all within 1 million $km^2$ (5.2%) of the observations, and generally produce root-mean-square (RMS) differences relative to TOMS/OMI of less than 0.9 million $km^2$ (14.6%). A notable exception to the latter is MERRA-2 with an RMS of 2.8 million $km^2$ (44.5%). This large RMS is attributable to an outlier year in 1994, when MERRA-2 had a very small ozone hole (Fig. 9). JRA-55 produces the smallest RMS difference relative to TOMS/OMI, while MERRA-2 model produces the smallest mean difference relative to these
observations.

MERRA did not produce an ozone hole in 1994, and produced very small ozone holes in 1993, 1997, 2009, and 2010. For related reasons, MERRA-2 did not produce an ozone hole in 1994, and produced a relatively small ozone hole in 1993. The elimination or reduction of the ozone hole during those years was caused by a lack of ozone observations for constraining the ozone field, as the processes that contribute to the development of the ozone hole are not represented in the parameterized
ozone chemistry used in MERRA and MERRA-2. In 1994, orbital drift of the *NOAA-11* satellite that provided the SBUV/2 TCO data assimilated by both MERRA and MERRA-2 led to a lack of ozone observations south of ~30°S during early Austral spring. *NOAA-11* SBUV/2 coverage was also limited in 1993. While both MERRA and MERRA-2 use *NOAA-11* SBUV, the version 8.6 data assimilated in the latter allowed less stringent quality screening criteria. Specifically, MERRA-2 uses observations made at solar zenith angles greater than 84°, excluded in MERRA, if they are otherwise marked as "good".
This results in a slightly better coverage of NOAA-11 SBUV in MERRA-2, explaining its better performance in 1993 and even 1994. The MERRA ozone hole was only weakly constrained by observations in late September 1997 because *NOAA-11* data only extended to 60°S–75°S between 21 September and 20 October. MERRA-2 does not have a low bias in ozone hole size during 1997 because it used data from *NOAA-14* rather than data from *NOAA-11*. The MERRA ozone hole was also





affected by orbital drift in the *NOAA-17* satellite and the concomitant loss of SBUV/2 observations at high southern latitudes during the austral springs in 2009 and 2010. MERRA-2 is unaffected during these years because of its assimilation of ozone observations from Aura OMI and MLS.

ERA-40 did not assimilate ozone data in 1989 and 1990. This resulted in a high bias in ozone concentrations and a very small ozone hole. The ERA-40 model also severely underestimated ozone hole area in 1997, most likely due to a gap in assimilated TCO from the Earthprobe TOMS instrument between August and December that year (Fig. 1; note that *NOAA-9* SBUV/2 profiles were assimilated during this timeframe as shown in Fig. 2). By contrast, the area of the ERA-Interim ozone hole was too large in 1995. This may be due to a lack of assimilated TCO observations in ERA-Interim during 1995 (Fig. 1).

**4.8 Long-term evolution of ozone**

Figure 10 shows the evolution of deseasonalized TCO anomalies from the reanalyses and assimilated observations from SBUV and TOMS/OMI. Also shown are the differences between the reanalyses and the primary TCO observations they assimilate. Both observational data sets show similar features, including a general trend toward decreasing ozone in the SH high latitudes, consistent with the Antarctic ozone hole depletion discussed in the previous section. However, in Fig. 10, comparison to the data set assimilated by a given reanalysis is done because differences between the TOMS/OMI and SBUV data sets show an apparent step change at the beginning of 2004. A comprehensive set of plots showing this step change, as well as reanalysis/observation differences separately for each data source, is provided in the supplementary material (Figs. S4-S5).

As expected, reanalyses agree more closely with TCO data that they assimilate than with data that they do not assimilate. For example, MERRA, MERRA-2, and CFSR assimilate SBUV data. The influence of SBUV on these reanalyses can be seen in the QBO-related anomalies in the tropics (particularly after ~1998) that are present in both the SBUV data and in the reanalyses that assimilate it. Differences between these reanalyses and SBUV are smaller in magnitude and more homogeneous in space and time than differences between these reanalyses and TOMS/OMI. The discontinuity in 2004 is particularly pronounced when MERRA and CFSR are compared against TOMS/OMI (Fig. S5). Similarly, differences between the ECMWF reanalyses and TOMS/OMI are generally more homogeneous and smaller in magnitude than differences between the ECMWF reanalyses and SBUV (Fig. S4). The period during which ERA-40 did not assimilate any ozone data (1989–1990) is also evident in Fig. 10. The stark contrast between this period and the surrounding years indicates the importance of data assimilation in constraining reanalysis ozone fields.

Figure 11 shows differences between reanalysis ozone fields and SWOOSH satellite limb profiler merged ozone data on two pressure levels (10 hPa and 70 hPa). This plot helps to evaluate disruptions in the temporal homogeneity of reanalysis ozone fields caused by changes in the assimilated observational data, and also provides a partially independent dataset for comparison with the reanalyses. The SWOOSH record is based primarily on v4.2 Aura MLS ozone starting in August 2004, so comparisons with reanalyses that assimilate MLS (i.e., MERRA-2 and ERA-Interim) after that time are not independent.





However, none of the observations used to construct the SWOOSH record prior to August 2004 were assimilated by these reanalyses.

At 10 hPa, CSFR, MERRA, and MERRA-2 show the best agreement with observations. At this level, ERA-Interim and JRA-25 have positive biases in both SH and NH midlatitudes, while JRA-55 has a negative bias relative to SWOOSH in the
tropics.

Overall, reanalysis ozone products do not exhibit large discontinuities at 10 hPa. As expected, both MERRA-2 and ERA-Interim show extremely good agreement with SWOOSH during the period in which they assimilate Aura MLS ozone data. Biases in these reanalyses undergo a step change when they start assimilating ozone profiles from Aura MLS ozone. For example, MERRA-2 assimilates Aura MLS data from August 2004 (Fig. 2), and at that time biases in 10 hPa ozone relative
to SWOOSH drop suddenly to less than 5% at all latitudes. This reduction is also apparent in ERA-Interim, which assimilates Aura MLS ozone data during 2008 and then from June 2009 through the present. Similar sudden reductions in ozone biases relative to SWOOSH are seen in ERA-Interim in both early 2008 and the latter half of 2009.

Differences between reanalysis ozone fields and SWOOSH are larger at 70 hPa. A strong discontinuity in the MERRA-2 time series occurs in mid-2004 when it begins to assimilate Aura MLS ozone data. To a lesser extent there is also a
discontinuity (in 2008 and again in mid-2009) when ERA-Interim begins assimilating Aura MLS ozone data. The large positive bias in MERRA-2 that starts in mid-2004 is also seen in comparisons to (non-assimilated) ozonesondes (Wargan et al., 2017). This positive bias is related to vertical averaging of the MLS data before assimilation by MERRA-2 (Wargan et al., 2017).

For the other reanalyses that don't assimilate MLS, there are generally not strong discontinuities that can be tied to
observing system changes. There does seem to be a change in the ERA-Interim differences at the beginning of 2003 when it begins to assimilate vertically resolved data from MIPAS and TCO from SCIAMACHY.  Beyond the discontinuities discussed above, at 70 hPa differences between the reanalysis ozone fields and SWOOSH are relatively consistent in time, with negative biases prevailing in CSFR, MERRA, and MERRA-2 (pre-Aura MLS), patchy biases in ERA-Interim, and mostly positive biases in JRA-25 and JRA-55 (especially in the tropics).

## 5 Evaluation of reanalysis water vapour

In this section, we evaluate reanalysis estimates of water vapour in and above the tropopause layer against available observations. In keeping with the S-RIP remit, this section focuses exclusively on evaluations of reanalysis water vapour products in the upper troposphere and stratosphere.

### 5.1 Zonal mean water vapour cross-sections

Figure 12 shows multi-annual zonal mean water vapour for 2005-2010 from the SDI MIM along with relative differences between each reanalysis and the MIM (calculated as $100*(R_i - MIM)/MIM$, where $R_i$ is a reanalysis field). In contrast to





ozone, the reanalyses do not consistently capture the zonal mean vertical distribution of water vapour. The pressure-level products provided by JRA-25 and JRA-55 do not include analysed stratospheric water vapour fields, while CFSR produces a stratosphere that is much too dry (low biases exceeding 60%). ERA-Interim, MERRA, and MERRA-2 show water vapour fields that are close to observations. These three systems resolve the distinct minimum in water vapour mixing ratios just

above the tropical tropopause, the second minimum in the lower stratosphere at SH high latitudes, and the increase in water vapour with increasing altitude. In contrast to other reanalyses, MERRA and MERRA-2 also extend up to the lower mesosphere (not shown), and, albeit with some limitations, they both capture the water vapour maximum found in the upper stratosphere (e.g., Hegglin et al., 2013), although slightly underestimated compared to observations, consistent with the simple parameterization as a 3-day relaxation to a climatology (Sects. 2.6 and 2.7).

CFSR is much too dry throughout the stratosphere and does not capture the typical structure of water vapour isopleths. This bias is due in part to the lack of assimilated observations to constrain the water vapour reanalyses at these altitudes and in part to the absence of a methane oxidation parameterization in the forecast model (Sect. 2.3). All reanalyses contain high biases relative to the SDI MIM at pressures greater than 100 hPa (see also Jiang et al., 2015), although this may be in part be explained by the increase in measurement uncertainty of satellite limb sounders with decreasing altitude in the upper

troposphere (Hegglin et al., 2013). Several studies have shown that Aura MLS contains a dry bias in the upper troposphere/lower stratosphere around 200 hPa (e.g., Davis et al., 2016; Vömel et al., 2007), and similarly a dry bias has been found in the upper troposphere for ACE-FTS (Hegglin et al., 2008).

### 5.2 Water vapour monthly mean vertical profiles and seasonal cycles

Figures 13 a and b show vertical profiles of water vapour for January (2005–2010 average) for the reanalyses and the SDI
MIM at two different latitudes 40°N and 70°S, respectively, along with the relative differences for each reanalysis with respect to the MIM. Figures 13c and d show the seasonal cycles of ozone for three different pressure levels at 40°N and 70°S, respectively. In general, the results shown reinforce the conclusions of the previous section.

The comparisons in Figs. 13 a and b reveal very good agreement (within ±10%) between ERA-Interim, MERRA, MERRA-2, and the observations at altitudes above 100 hPa. As mentioned in the previous section, water vapour from CSFR
is unrealistic in the stratosphere, with values much lower than those observed. The reanalyses show large inconsistencies between their absolute values at altitudes below 100 hPa, leading to sharp increases in their relative differences with respect to the MIM of >100%. These relative differences are systematically positive except for in CFSR and JRA-25, pointing towards potential negative biases in the water vapour observations at these altitudes (e.g., Hegglin et al., 2013). The results may also indicate that the reanalyses produce an excessively moist tropical upper troposphere and/or excessive mixing of
moist tropospheric air into the extratropical lowermost stratosphere. The 100 hPa level is one of the most important levels for stratospheric water vapour studies, because it is near the level where stratospheric water vapour entry mixing ratios are set in the tropics (Fueglistaler et al., 2009) and because it is near the peak region of the radiative kernel for water vapour in the extratropics (Gettelman et al., 2011).





The agreement between the reanalyses and observations varies by month, as shown in Figs. 13 c and d for selected pressure levels (250, 100, and 50 hPa) and latitude bands (30°N-50°N and 60°S-80°S). At NH mid-latitudes (30°N-50°N; Fig. 13c) at 250 hPa, all reanalyses are biased high relative to the observations by more than 100%, lending further support to the results by Jiang et al. (2015), who compared the reanalyses to Aura MLS alone, which is known to have a low bias

around this altitude (Davis et al., 2016; Hegglin et al., 2013; Vömel et al., 2007). JRA-25 and JRA-55 have the smallest high biases relative to observations at 250 hPa. At 100 hPa and 50 hPa, ERA-Interim, MERRA, and MERRA-2 perform best, with approximately correct mean values, but somewhat underestimated seasonal cycle amplitudes. As noted earlier, a significant portion of the agreement in MERRA and MERRA-2 results from the relaxation of stratospheric water vapour towards a climatology that is based in part on Aura MLS data (which are also included in the SDI MIM). JRA-55 (JRA-25)

has mean values that are much too large (small) at 100 hPa. In addition to being too dry at 100 and 50 hPa, CSFR also has incorrect amplitude and phase of the seasonal cycle at these levels.

At SH high latitudes (60°S-80°S; Fig. 13d), all reanalyses show approximately the right phase, but overestimate mean values and amplitudes at 250 hPa, similar to the results at NH mid-latitudes. At 100 and 50 hPa, ERA-Interim captures the phase and amplitude of the observed seasonal cycle best when compared to the other reanalyses, but exhibits a slight low

bias at 50 hPa. MERRA and MERRA-2 show also quite good agreement in terms of mean value, amplitude, and phase at 100 hPa, but overestimate mean values at 50 hPa, and also show a somewhat early minimum followed by an increase in September that occurs about a month earlier than observations. JRA-25 somewhat underestimates the mean value, but shows a similar phase and amplitude as the observations at 100 hPa. JRA-55 on the other hand, strongly overestimates the amplitude of the seasonal cycle at this level with mean values that are much too high. CSFR shows too low values at both

100 and 50 hPa, but captures the seasonality somewhat better than it does in the NH mid-latitudes.

**5.3 Interannual variability in water vapour**

Figure 14 shows time series of interannual variability in water vapour and its anomalies based on observations and reanalysis products during 2005–2010. At 250 hPa in NH midlatitudes (40°N-60°N), the reanalyses generally follow the observed interannual variability extremely well, especially JRA-25, JRA-55, and MERRA. CSFR seems to exhibit an

underlying positive trend in its timeseries that is stronger than that observed. And as noted previously, all reanalyses are wetter than observations at this level by approximately a factor of two.

At 100 hPa in the tropics (a level that is often used to estimate stratospheric water vapour entry mixing ratios), all reanalyses except CSFR compare reasonably well with the observed anomalies. Perhaps surprisingly, JRA-25 captures the interannual anomalies quite well despite being biased in its seasonal cycle. CSFR shows no clear interannual variability and

produces water vapour mean values as low as 0 ppmv. CSFR begins to produce more realistic water vapour concentrations at these levels in 2010 with values that are larger and in better agreement with observations than those in the other reanalyses. This change is discussed further in Sect. 5.4 Note that the SDI MIM for this level only includes Aura-MLS and ACE-FTS due to known problems in SCIAMACHY and MIPAS data in this region (Hegglin et al., 2013).





At 50 hPa in the SH high latitudes (60°S-80°S), MERRA and MERRA-2 have roughly correct water vapour mean values, whereas CFSR and ERA-Interim are too low. MERRA and MERRA-2 both place the minimum during austral winter (from dehydration processes in the cold polar vortex) about one month too early. Except for CFSR, the other reanalyses capture the correct structure in the interannual variability, including the prominent positive anomaly in 2010. MERRA and MERRA-2

show less variability than observed, which is unsurprising given their strong relaxation to the climatology.

### 5.4 Tropical tape recorder in water vapour

Representations of the tropical tape recorder (Mote et al., 1996) provide an additional illustration of problems in reanalysis stratospheric water vapour products. Figure 15 shows the time–height evolution of water vapour in reanalyses and the

merged SWOOSH observations averaged over the 15°S–15°N tropical band. Anomalies are calculated separately for each data set, relative to the mean seasonal cycle at each level for the period 1992-2014 (except ERA-40, which is 1992-2002), when all reanalyses (except ERA-40) overlap. Variations in these fields reflect changes in the mixing ratio of water vapour entering the tropical lower stratosphere, as driven by variations in tropical tropopause temperatures and the subsequent vertical propagation in the ascending branch of the stratospheric overturning circulation. Interannual variability in both water

vapour entry mixing ratios and ascent rate (the vertical slope of the signal) is superimposed on this mean seasonal cycle. Although reanalyses do not reproduce observed water vapour concentrations in the stratosphere, most reanalyses do produce a tropical tape recorder signal.

As previously discussed, CFSR (Fig. 15a) produces water vapour concentrations near zero in the stratosphere for most of the record, although unrealistically wet values appear above 20 hPa at certain times (e.g. 1995 and 1999). These upper

stratospheric wet anomalies (and several others that occurred before 1992) all correspond to transitions in the main CFSR production stream (see Fig. 2, Fujiwara et al., 2017). We hypothesize that these wet anomalies are a remnant of a wet bias in the model initialisation that remains after the ~1-year spinup. Additional step changes in water vapour are evident at the beginning of 2010 and at the beginning of 2011. The latter step change corresponds to the transition from CFSR (CDAS-T382) to CFSv2 (CDAS-T574) at the beginning of 2011. As discussed in Sect. 2.2, CFSv2 is intended as a continuation of

CFSR but has differences in model resolution and physics relative to the original system. Although the reasons for the step change at the beginning of 2010 are not known definitively, we note that CFSR was extended for the year 2010 following its original completion over the 1979-2009 time period. This extension used the original CDAS-T382 system but with some slight changes to the forecast model. It is likely that the CFSR 2010 run was performed without a sufficiently long spin-up period, or that a change to the model configuration resulted in the observed water vapour discontinuity beginning in 2010.

ERA-40 and ERA-Interim (Fig. 15 c, e) are generally drier than the SWOOSH observations (Fig. 15 k), although the ERA-Interim represents an evident improvement over ERA-40 in this respect. Both MERRA and MERRA-2 (Fig. 15 g, i) are close in magnitude to SWOOSH, but this agreement is expected given that both systems relax stratospheric water vapour to a climatology based on Aura MLS and HALOE (Sect. 2.6, 2.7).





The reanalyses all produce tape recorder slopes that are more vertical than suggested by the observations, indicating that vertical upwelling in the tropical stratosphere is too strong in reanalyses. Although biases and differences in tropical stratospheric upwelling have been addressed quantitatively for a subset of reanalyses elsewhere (Abalos et al., 2015; Jiang et al., 2015), the SWOOSH data shown in Fig. 15 enable a comparison that extends beyond the Aura MLS record. This

extension allows for comparison to ERA-40, and shows that ERA-Interim benefits from a much-improved representation of stratospheric water vapour and its variability relative to its predecessor.

Figure 15 also shows interannual variability in tropical stratospheric water vapour as represented by the anomaly from the mean seasonal cycle at each level. Interannual variability in the tape recorder signal is related to interannual variability in cold-point tropopause temperatures (Fig. 15 m), with warm anomalies at the tropopause corresponding to wet anomalies in

the tape recorder and vice versa. Although the reanalyses produce almost identical interannual variations in tropical tropopause temperatures over the period considered here, their interannual variations in stratospheric water vapour differ substantially. The strong relaxation to climatology applied in MERRA and MERRA-2 results in very little interannual variability above 60 hPa because of the short nudging timescale for WV (3 days). ERA-40 produces a very large wet anomaly during the 1997–1998 El Niño that coherently propagates upwards. This anomaly is wetter than that suggested by

SWOOSH and the other reanalyses. SWOOSH and the reanalyses all show a wet anomaly near 100 hPa in the tropics during the 1997–1998 El Niño, but this anomaly does not correspond to a strong warm excursion in cold-point temperature.

Randel et al. (2006) reported the occurrence of a sudden drop in stratospheric water vapour that persisted for ~5 years during the early 2000s. This drop is evident in the cold-point temperature and SWOOSH water vapour anomalies (Fig. 15 l, m). The reanalyses generally capture the drop in stratospheric WV around 2000, with the caveat that the relaxation to a

monthly mean climatology in MERRA and MERRA-2 damps the associated signals above the lowermost stratosphere.

## 6 Conclusions

In this paper, we described the basic treatment of ozone and water vapour in reanalyses, and presented comparisons both among reanalyses and between reanalyses and observations (both assimilated and independent). Here we briefly summarize the most influential characteristics and differences in the treatment of ozone and water vapour in reanalyses along with the

key results of the intercomparisons.

The treatment of ozone and water vapour varies substantially among reanalyses. Some reanalyses prescribe ozone climatologies and do not treat ozone prognostically (R1, R2), some reanalyses specify ozone as a boundary condition generated by an offline chemical transport model (JRA-25, JRA-55), and some reanalyses treat ozone as a prognostic variable with parameterized photochemical production and loss (CFSR, ERA-40, ERA-Interim, MERRA, MERRA-2). Only

ERA-40 and ERA-Interim contain a parameterization of heterogeneous ozone loss processes.

The reanalyses also assimilate different sets of ozone observations, with generally similar observation usage for reanalyses produced by the same reanalysis centre. All reanalyses that assimilate ozone observations rely heavily on total column ozone



observations from some combination of satellites carrying the TOMS and SBUV sensors. Several recent reanalyses (including MERRA-2 and ERA-Interim) use the newest generation of vertically resolved ozone measurements (e.g., Aura MLS).

Reanalyses all assimilate tropospheric humidity information via some combination of radiosondes, satellite radiances, GNSS-RO bending angles, and retrievals of atmospheric hydrological quantities (e.g., total column water vapour or rain rate). None of the reanalyses assimilate WV observations in the stratosphere, although information from tropospheric observations may propagate upward in some systems. Beyond these similarities, the treatment of stratospheric water vapour varies substantially among the reanalyses. For example, the specific cut-off altitude up to which radiosonde humidity data are assimilated varies from one reanalysis to another, using either a fixed pressure level or the diagnosed tropopause. ERA-

40 and ERA-Interim are the only reanalyses that include a water vapour source from methane oxidation. MERRA and MERRA-2 relax their fields to a water vapour climatology based on satellite observations (e.g., including Aura MLS), while other reanalyses simply do not provide valid data in the stratosphere (e.g., CSFR, JRA-25, JRA-55, R1, R2). These latter reanalyses prescribe a climatology or constant value for stratospheric water vapour as input to the forecast model radiative transfer code.

Given these differences amongst reanalysis treatments of ozone and WV, it is perhaps unsurprising that comparisons between reanalyses and observations also vary widely. Comparisons against assimilated observations of total column ozone (TCO) show that reanalyses generally reproduce TCO well, within ~10 DU (~3%). Key limitations that result in larger errors and uncertainties include a general lack of TCO data during polar night and the absence of heterogeneous chemistry from most reanalysis ozone schemes (except in ERA-40 and ERA-interim where it is introduced as a simple parameterization

activated when the local temperature falls below 195K). The vertical distributions of stratospheric ozone and WV in reanalyses are unconstrained by observations through most of the record, owing to vertically-resolved data generally not being used in the assimilation systems. The situation for ozone is slightly better than that for WV, because stratospheric ozone observations are assimilated and because the ozone parameterizations are more advanced.

    From the middle to upper stratosphere, reanalysis ozone profiles are within ±20% of observations from the SPARC Data

Initiative, although the comparisons are not truly independent for MERRA-2 or ERA-Interim because they assimilate data from Aura MLS, one of the instruments that contribute to the SPARC Data Initiative dataset. In the upper troposphere and lower stratosphere, biases increase to ±50% for ozone.

    MERRA-2 performs particularly well for ozone through much of the stratosphere. This is mainly due to the assimilation of the vertically resolved Aura MLS observations, which have helped to address difficulties in reproducing vertical

distributions of ozone, particularly during polar night; however, these data are only available since late 2004 and are only assimilated by a few reanalyses. The use of reanalysis ozone for Antarctic ozone hole studies is therefore problematic. The reanalyses produce reasonable ozone holes when observations are available, but the timing and area of reanalysis ozone holes is highly biased when observations are unavailable. Also, apart from JRA-55, most reanalyses seem to exhibit a drift in the extent of the ozone hole area when compared to TOMS/OMI observations.





None of the reanalyses assimilate observations of stratospheric water vapour, resulting in large differences between reanalyses and independent observations. CFSR has an extreme dry bias in the stratosphere through 2009, with monthly mean values often approaching 0 ppmv. Although MERRA and MERRA-2 produce reasonable values for stratospheric water vapour, these values represent a strong relaxation to a fixed annual climatology at pressures less than 50 hPa. Hence,

mid- and upper-stratospheric water vapour does not undergo physically meaningful variations in MERRA or MERRA-2. ERA-40 and ERA-Interim produce a true "prognostic" water vapour field in the stratosphere. ERA-Interim produces surprisingly reasonable values given that its field is predominantly controlled by dehydration in the TTL and a very simple parameterization of methane oxidation. In the upper troposphere and lower stratosphere, reanalyses are around a factor of two wetter than the SPARC Data Initiative instruments used here, although the observations also have relatively large

disagreements in this region.

Because of the lack of assimilated observations and the deficiencies in representation of the relevant physical processes, we recommend that reanalysis stratospheric water vapour fields should generally not be used for scientific data analysis, and stress that any examination of these fields must account for their inherent limitations and uncertainties. Future efforts toward the collection and assimilation of observational data with sensitivity to stratospheric water vapour, the reduction of

reanalysis temperature biases in the TTL, and improvements in the representation of processes that control the entry mixing ratios or subsequent evolution of water vapour in the stratosphere could facilitate more reliable stratospheric water vapour fields in reanalyses.

**Code availability**

Code for creating the common-grid data files and plots are available from the corresponding author upon request.

**Data availability**

The reanalysis data files necessary to create the "common grid" data files used here are available through the CREATE project website (https://cds.nccs.nasa.gov/tools-services/create/). Reanalysis total column ozone data was downloaded from the NCAR RDA (https://rda.ucar.edu/). SBUV data are available at https://acd-ext.gsfc.nasa.gov/Data_services/merged/. TOMS/OMI data are available at https://disc.gsfc.nasa.gov/Aura/data-holdings/OMI/omto3d_v003.shtml. SPARC DI data

are available at http://www.sparc-climate.org/data-center/data-access/sparc-data-initiative/. Aura MLS satellite data are available at https://disc.sci.gsfc.nasa.gov/Aura/data-holdings/MLS. SWOOSH data are available at https://www.esrl.noaa.gov/csd/swoosh/.



**Appendix A**

Major abbreviations and terms are defined below.

1D-Var: 1-dimensional variational data assimilation scheme

MIM     Multi-Instrument Mean

20CR: 20th Century Reanalysis of NOAA and CIRES

AIRS: Atmospheric Infrared Sounder

Aqua: a satellite in the EOS A-Train satellite constellation

ATMS: Advanced Technology Microwave Sounder

ATOVS: Advanced TIROS Operational Vertical Sounder

Aura: a satellite in the EOS A-Train satellite constellation

CDAS: Climate Data Assimilation System

CFC: chlorofluorocarbon

CFSR: Climate Forecast System Reanalysis of NCEP

CFSv2: Climate Forecast System, version 2

CHEM2D: The NRL 2-Dimensional photochemical model

CHEM2D-OPP: CHEM2D Ozone Photochemistry Parameterization

CIRES: Cooperative Institute for Research in Environmental Sciences (NOAA and University of Colorado Boulder)

CREATE: Collaborative REAnalysis Technical Environment

CrIS: Cross-track Infrared Sounder

CTM: chemical transport model

ECMWF: European Centre for Medium-Range Weather Forecasts

EOS: NASA's Earth Observing System

ERA-15: ECMWF 15-year reanalysis

ERA-20C: ECMWF 20th century reanalysis

ERA-40: ECMWF 40-year reanalysis

ERA5: A forthcoming reanalysis developed by ECMWF

ERA-Interim: ECMWF interim reanalysis

GFS: Global Forecast System of the NCEP

GNNS-RO: Global Navigation Satellite System Radio Occultation (see also GPS-RO)

GPS-RO: Global Positioning System Radio Occultation (see also GNSS-RO)

HIRS: High-resolution Infrared Radiation Sounder

IASI: Infrared Atmospheric Sounding Interferometer

IFS: Integrated Forecast System of the ECMWF





IR: Infrared

JCDAS: JMA Climate Data Assimilation System

JMA: Japan Meteorological Agency

JRA-25: Japanese 25-year Reanalysis

JRA-55: Japanese 55-year Reanalysis

JRA-55AMIP: Japanese 55-year Reanalysis based on AMIP-type simulations

JRA-55C: Japanese 55-year Reanalysis assimilating Conventional observations only

MERRA: Modern Era Retrospective-Analysis for Research

MIM: Multi Instrument Mean

MIPAS: Michelson Interferometer for Passive Atmospheric Sounding

MLS: Microwave Limb Sounder

MRI-CCM1: Meteorological Research Institute (JMA) Chemistry Climate Model, version 1

MSU: Microwave Sounding Unit

NASA: National Aeronautics and Space Administration

NCAR: National Center for Atmospheric Research

NCEP: National Centers for Environmental Prediction of the NOAA

NMC: National Meteorological Center (now NCEP)

NOAA: National Oceanic and Atmospheric Administration

NRL: Naval Research Laboratory

ODS: Ozone Depleting Substance

OMI: Ozone Monitoring Instrument

QBO: quasi-biennial oscillation

R1: NCEP-NCAR Reanalysis 1

R2: NCEP-DOE Reanalysis 2

RDA: Research Data Archive

RH: Relative Humidity

RTTOV: Radiative Transfer for TOVS

SEVIRI: Spinning Enhanced Visible and InfraRed Imager

SBUV & SBUV/2: Solar Backscatter Ultraviolet Radiometer

SCIAMACHY: Scanning Imaging Absorption Spectrometer for Atmospheric Chartography

SDI: SPARC Data Initiative

SPARC: Stratosphere-troposphere Processes And their Role in Climate

S-RIP: SPARC Reanalysis Intercomparison Project

SSM/I or SSMI: Special Sensor Microwave Imager

SSU: Stratospheric Sounding Unit

TCWV: Total Column Water Vapor

TCO: Total Column Ozone

TIROS: Television Infrared Observation Satellite

5 TMI: Tropical Rainfall Measuring Mission (TRMM) Microwave Imager

TOA: top of atmosphere

TOMS: Total Ozone Mapping Spectrometer

TOVS: TIROS Operational Vertical Sounder

TTL: Tropical tropopause layer

10 UV: Ultraviolet

VTPR: Vertical Temperature Profile Radiometer

WV: Water Vapor



*Acknowledgements*. We thank the World Climate Research Programme and SPARC for supporting the S-RIP project. The authors are grateful to both past and present SPARC co-chairs Ted Shepherd, Greg Bodeker, Joan Alexander, and Neil Harris for their support and encouragement of S-RIP. The Information Initiative Center of Hokkaido University, Japan has hosted the S-RIP web server since 2014. We thank the reanalysis centres for providing their support and data products. The

5    British Atmospheric Data Centre (BADC) of the UK Centre for Environmental Data Analysis (CEDA) has provided a virtual machine for data processing and a group workspace for data storage. We thank Peter Haynes, Gabriele Stiller, and William Lahoz for serving as the editors for the special issue "The SPARC Reanalysis Intercomparison Project (S-RIP)" in this journal. We thank Karen Rosenlof for valuable comments and suggestions on this manuscript. MF's contribution was financially supported in part by the Japan Society for the Promotion of Science (JSPS) through Grants-in-Aid for Scientific

10   Research (26287117 and 16K05548). Work at the Jet Propulsion Laboratory, California Institute of Technology, was carried out under a contract with the National Aeronautics and Space Administration.





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



**Table 1.** Key characteristics of ozone treatment in reanalyses

| Reanalysis | Primary TCO data sources | Vertical profile data sources | Stratospheric $O_3$ used in radiative transfer | Stratospheric $O_3$ treatment | Photochemical parameterization |
|---|---|---|---|---|---|
| NCEP R1 | None | None | Climatology | None | None |
| NCEP R2 | None | None | Climatology | None | None |
| CFSR | SBUV | SBUV | Analysed | Prognostic | CHEM2D-OPP |
| ERA-40 | TOMS | SBUV | Climatology | Prognostic | CD86 |
| ERA-I | Same as ERA-40 | SBUV, GOME, MLS, MIPAS | Same as ERA-40 | Same as ERA-40 | Same as ERA-40 |
| JRA-25 | TOMS (1979–2004)[a] OMI (2004–) | Nudging to climatological profile | Daily values from offline CTM | Daily values from offline CTM | Shibata et al. (2005) |
| JRA-55 | Same as JRA-25 | None | Daily values from updated offline CTM | Daily values from updated offline CTM | Shibata et al. (2005) |
| MERRA | SBUV | SBUV | Analysed | Prognostic | Stajner et al. (2008) |
| MERRA-2 | SBUV (1980–9/2004) OMI (9/2004–) | SBUV, MLS | Same as MERRA | Same as MERRA | Same as MERRA |

[a] Offline CCM nudged to TOMS/OMI data.




**Table 2.** Key characteristics of water vapour treatment in reanalyses

| Reanalysis | Assimilation of satellite humidity radiances? | Highest level of assimilated WV observations | Highest level of analyzed WV[1] | Stratospheric WV used in radiative transfer | Stratospheric WV treatment | Stratospheric methane oxidation parameterization? |
|---|---|---|---|---|---|---|
| NCEP R1 | No | 300 hPa | 300 hPa | Climatology | None | No |
| NCEP R2 | No | 300 hPa | 10 hPa (RH only) | Climatology | None | No |
| CFSR | Yes | 250 hPa | None | Analysed; negative values set to 0.1 ppmv | Prognostic | No |
| ERA-40 | Yes | Diagnosed tropopause. Radiosonde humidity generally used to 300 hPa | Diagnosed tropopause | Analysed | Prognostic | Yes. Relaxation to 6 ppmv WV at stratopause |
| ERA-I | Yes | Same as ERA-40 | Diagnosed tropopause | Analysed | Prognostic | Yes. Relaxation to 6.8 ppmv WV at stratopause |
| JRA-25 | Yes | 100 hPa | 50 hPa | Constant 2.5 ppmv | Prognostic[2] | No |
| JRA-55 | Yes | 100 hPa | 5 hPa | Climatological annual mean from HALOE and UARS MLS during 1991–1997 | Prognostic[2] | No |
| MERRA | Yes | 300 hPa | None | Analysed | 3-day relaxation to zonal-mean monthly-mean satellite-based climatology | No |
| MERRA-2 | Yes | 300 hPa | None | Same as MERRA | Same as MERRA | No |

[1] Level above which assimilation-related increments are not allowed.

[2] Water vapour not provided above 100 hPa in pressure level analysis products.




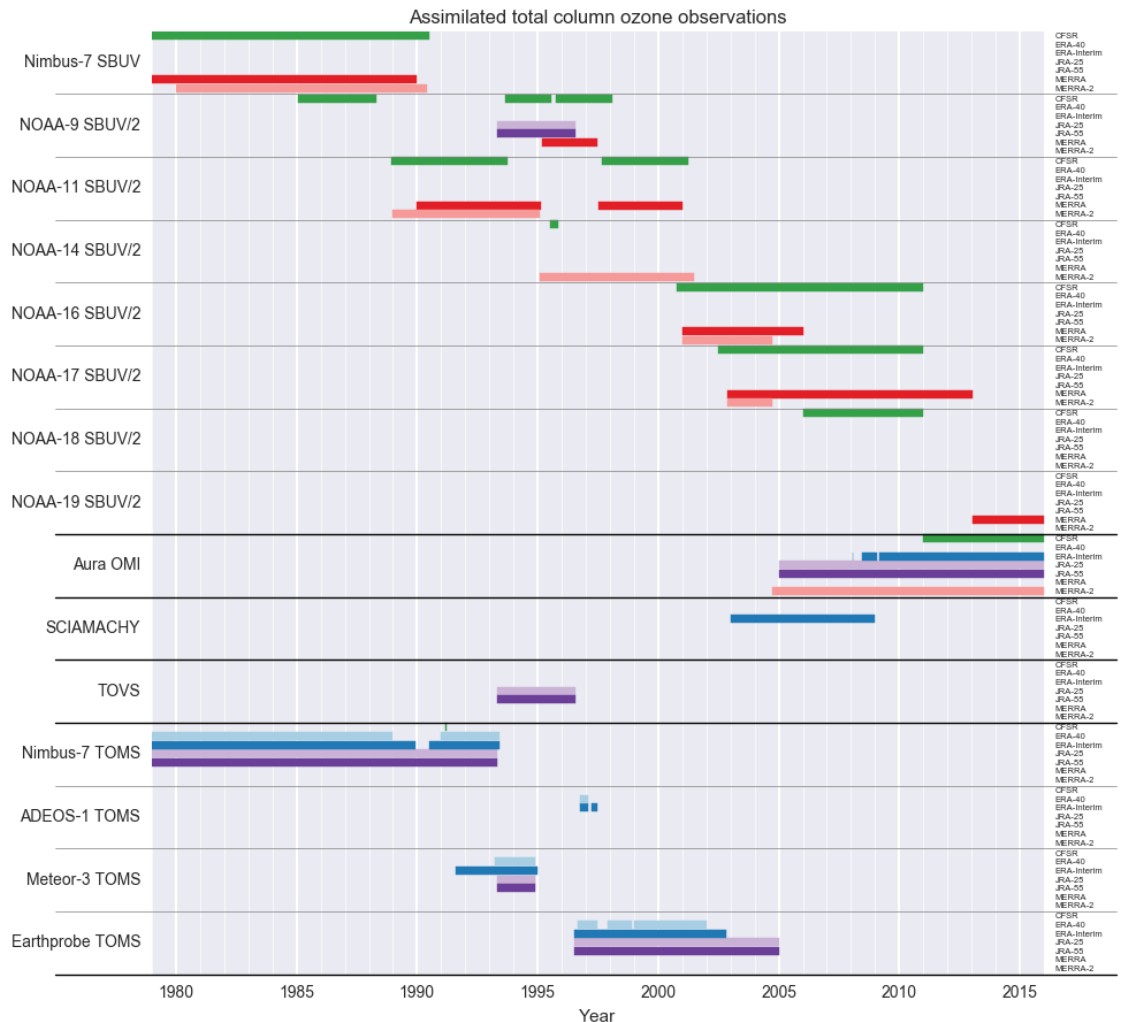

**Figure 1:** Total column ozone data by instrument as assimilated by the different reanalyses.





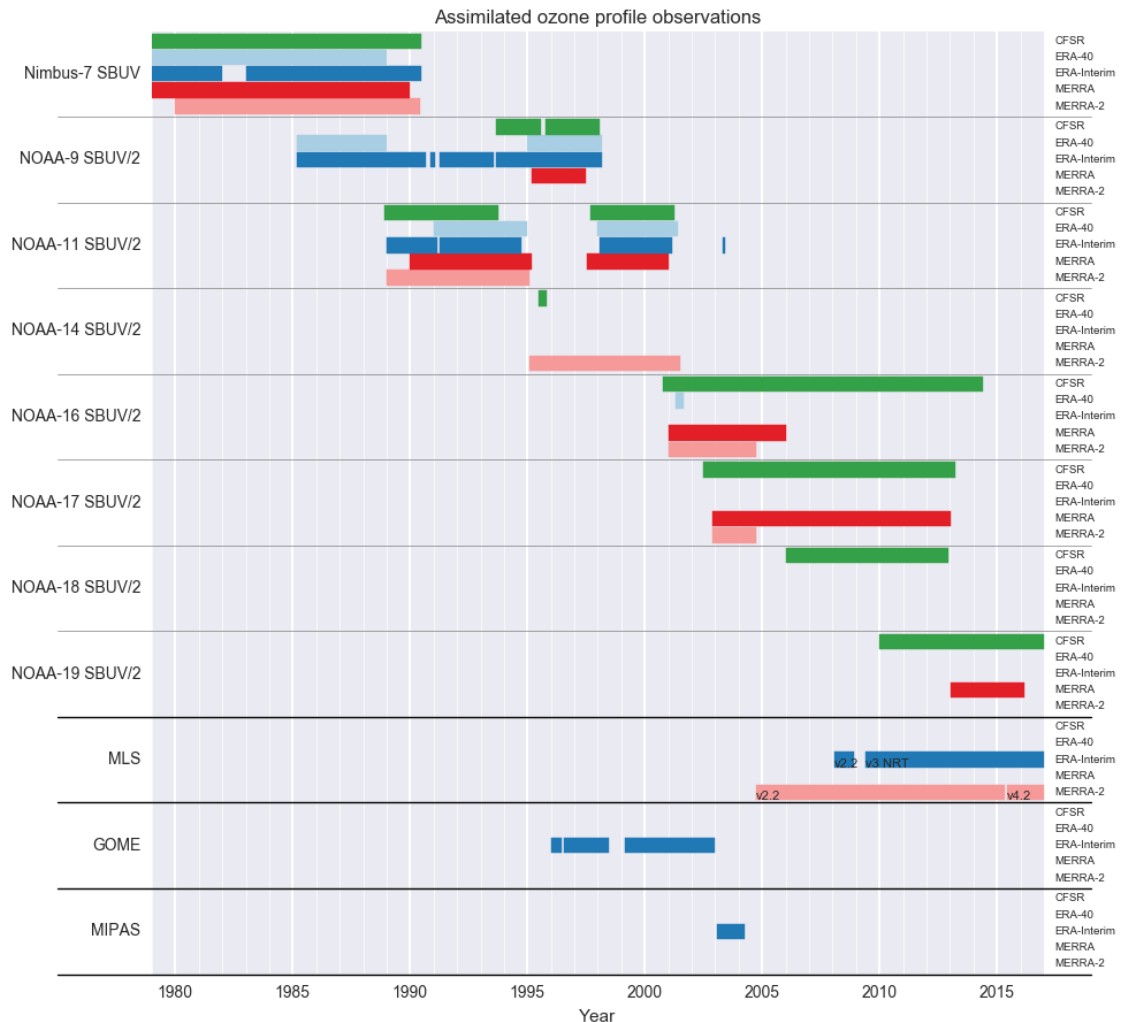

**Figure 2:** Ozone vertical profile observations by instrument as assimilated by the different reanalyses.


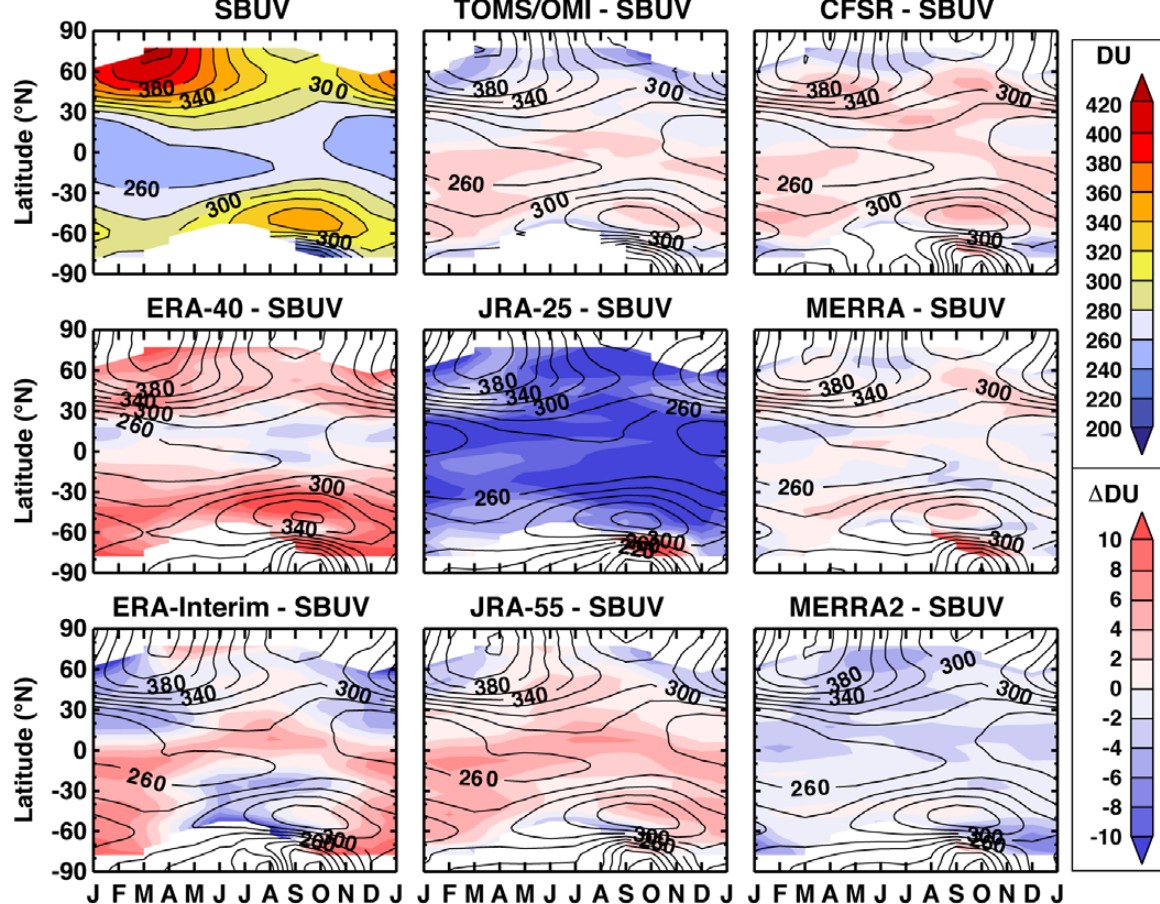

**Figure 3:** Zonal- and monthly-mean total column ozone climatology over 1981–2010 from SBUV observations (uppermost left panel), along with the absolute differences between each reanalysis and SBUV. The difference between TOMS/OMI and SBUV is also shown (uppermost middle panel). Line contours show each reanalysis' respective climatology. Both climatology and observational reference to calculate differences for ERA-40 are for the time period Jan 1981–Aug 2002 in order to avoid sampling issues.





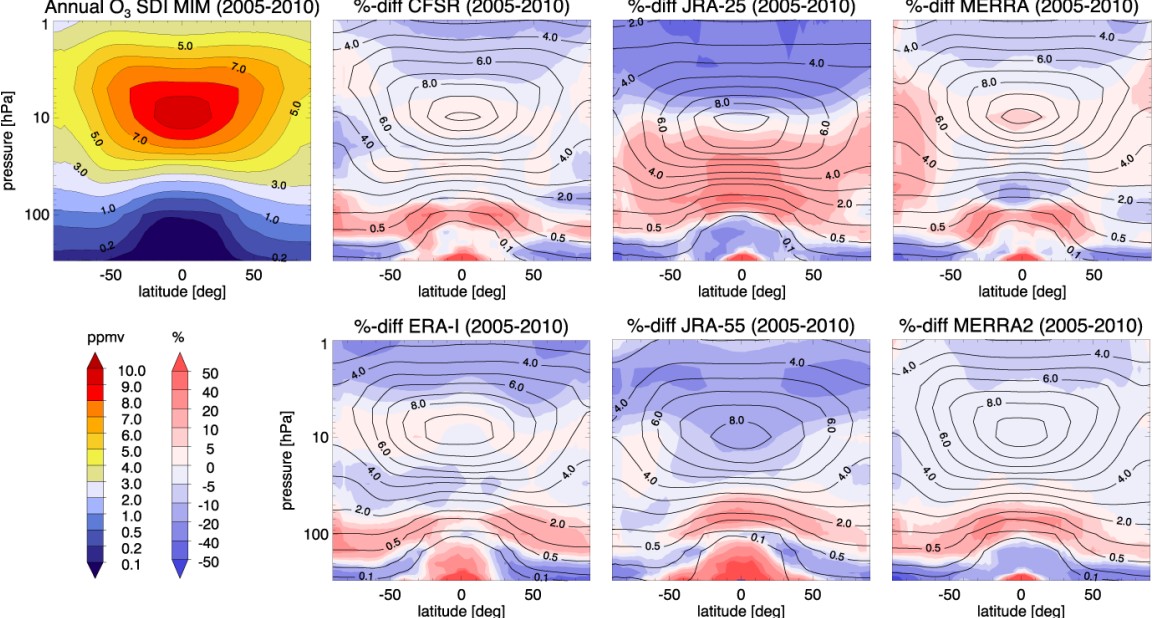

**Figure 4:** Multi-annual zonal mean ozone cross sections averaged over 2005–2010 for the SPARC Data Initiative multi-instrument mean (SDI MIM) (upper left), along with the relative differences between reanalyses and observations as $(R_i–MIM)/MIM*100$, where $R_i$ is a reanalysis field. Also shown in contours are the respective zonal mean climatologies for the different reanalyses.



**Figure 5:** Multi-annual mean vertical ozone profiles over 2005–2010 for January at **(a)** 40N and **(b)** 70S from the SPARC Data Initiative multi-instrument mean (SDI MIM) (black) and the six reanalyses (coloured). Absolute values are shown in the left and relative differences in the right panels for each comparison. Relative differences are calculated as $(R_i-MIM)/MIM*100$, where $R_i$ is a reanalysis profile. Black dashed lines provide the ±1-sigma uncertainty (as calculated by the standard deviation over all instruments and years available) in the observational mean. Horizontal dashed lines in grey indicate the pressure levels (150, 50, and 10 hPa) for which seasonal cycles are shown in panels **(c)** and **(d)** for the two latitude ranges 30-50N and 60-80S, respectively.



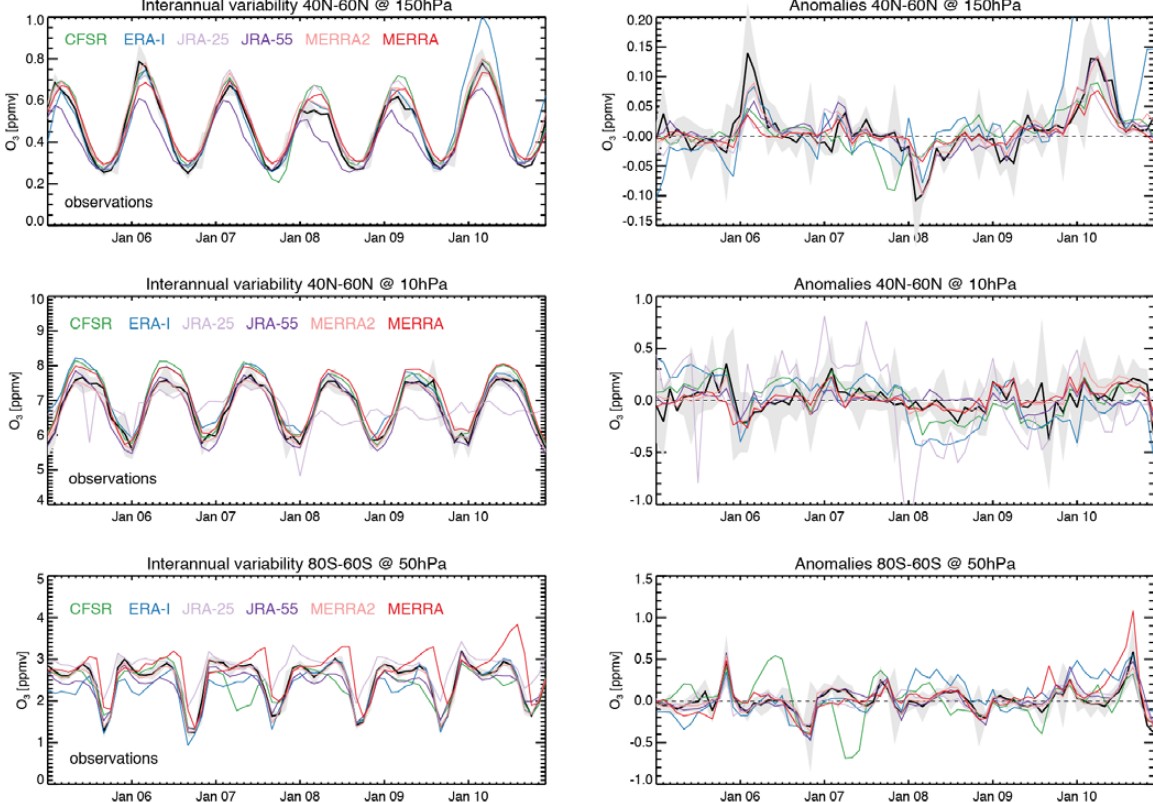

**Figure 6:** Interannual variability (left column) and deseasonalized anomalies (right column) for ozone during 2005–2010 for the SPARC Data Initiative multi-instrument mean (SDI MIM, black) and the six reanalyses (coloured). Results are shown for three different pressure levels and latitude ranges (top to bottom: 50 hPa at 60–80°S, 10 hPa at 20°S–20°N, and 100 hPa at 40–60°N). Grey shading indicates observational uncertainty (±1-sigma) calculated as the standard deviation over all instruments and years available.





**Figure 7:** Comparison of the equivalent latitude–time evolution of each reanalysis ozone field and MLS on the 520 K isentropic surface (~50 hPa; ~20 km altitude) during the Aura mission September 2004 – December 2013. (Left) Mixing ratios (ppmv) for MLS and the reanalyses MERRA, MERRA-2, ERA-Interim, CFSR, and JRA-55 (top to bottom). (Right) differences (ppmv) between each reanalysis and MLS ($R_i$ – MLS). Overlays are scaled potential vorticity (Manney et al., 1994) contours of 1.4 and 1.6 x $10^{-4}$ $s^{-1}$ from the corresponding reanalysis, which are intended to represent the wintertime polar vortex edge. Dynamical fields for the MLS panel are from MERRA.



**Figure 8:** QBO ozone signal from the SPARC Data Initiative observations (upper left) during 2005–2010, defined as altitude–time cross-sections of deseasonalized ozone anomalies averaged over the 10°S–10°N tropical band. Observations are based on three satellite data sets. The other panels show the differences in QBO ozone signals between each reanalysis and the observations ($R_i$–MIM) with the black contours showing the QBO ozone signal generated by each corresponding reanalysis.





**Figure 9: (a)** Ozone hole mean area calculated from TOMS/OMI observations and the reanalyses for 21 September through 20 October of 1981–2010. **(b)** Differences between ozone hole mean areas from reanalyses and TOMS/OMI observations ($R_i$ – observed).





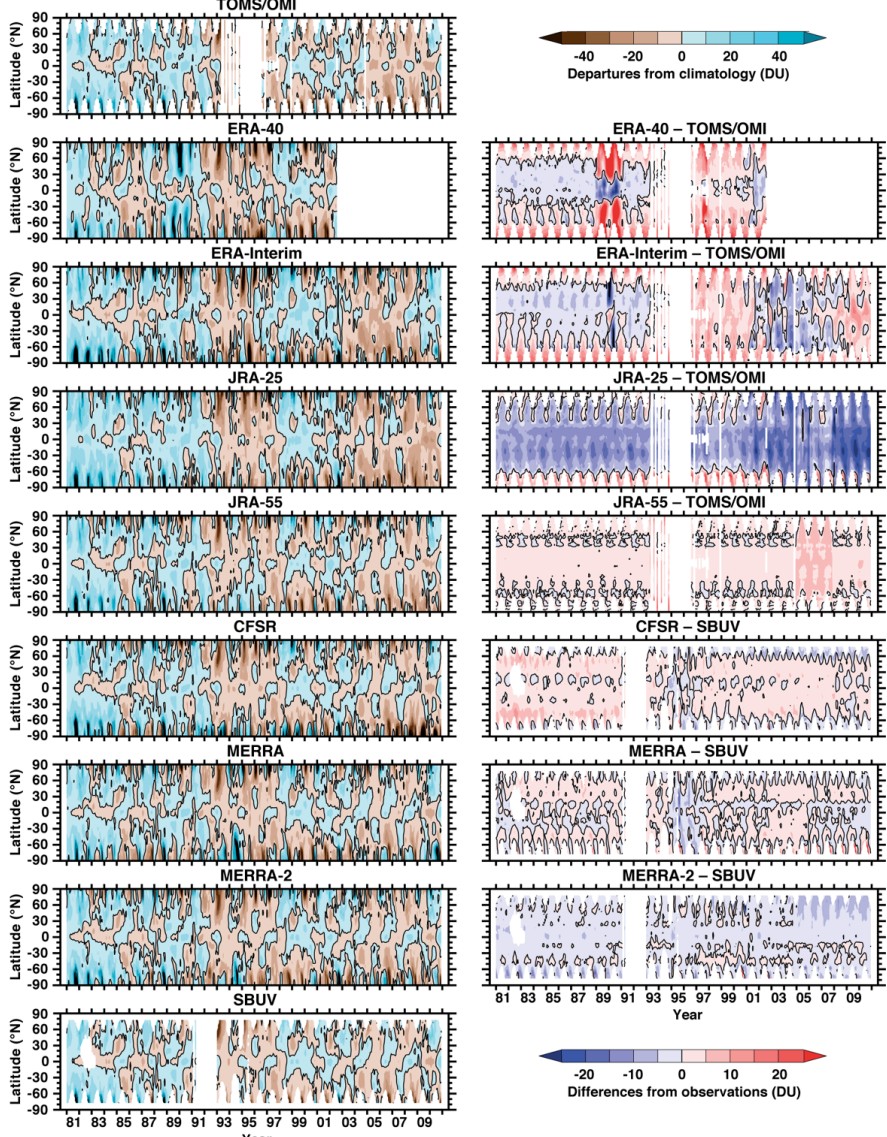

**Figure 10: Departures of TCO from the zonal- and monthly-mean 1981-2010 climatology for TOMS/OMI (left column, top row), SBUV (left column, bottom row), and reanalyses (left column, other rows). (Right column) Differences between reanalyses zonal- and monthly-mean TCO and the primary TCO observations that they assimilate. The black contour is at 0 DU.**





**Figure 11:** Latitude–time evolution of relative differences between ozone reanalyses and the merged SWOOSH ozone record at 10 hPa and 70 hPa. White indicates missing data, and light grey indicates near-zero differences (e.g., between MERRA2 and SWOOSH after mid-2004).



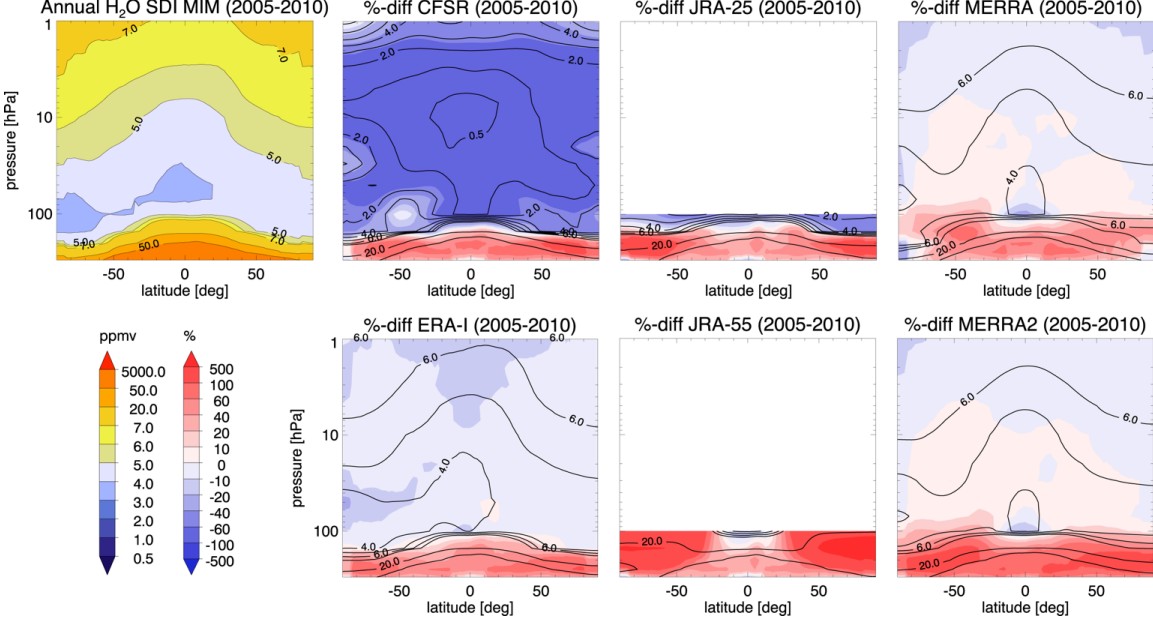

**Figure 12:** Multi-annual zonal mean water vapour cross sections averaged over 2005–2010 for the SPARC Data Initiative multi-instrument mean (SDI MIM) (upper left), along with the relative differences between reanalyses and observations as ($R_i$–MIM)/MIM*100, where $R_i$ is a reanalysis field. Also shown in contours are the respective zonal mean climatologies for the different reanalyses.





**Figure 13:** Multi-annual mean vertical water vapour profiles over 2005–2010 for January at **(a)** 40N and **(b)** 70S from the SPARC Data Initiative multi-instrument mean (SDI MIM) (black) and the six reanalyses (coloured). Absolute values are shown in the left and relative differences in the right panels for each comparison. Relative differences are calculated as (R$_i$–MIM)/MIM*100, where R$_i$ is a reanalysis profile. Black dashed lines provide the ±1-sigma uncertainty (as calculated by the standard deviation over all instruments and years available) in the observational mean. Horizontal dashed lines in grey indicate the pressure levels (250, 100, and 50 hPa) for which seasonal cycles are shown in panels **(c)** and **(d)** for the two latitude ranges 30-50N and 60-80S, respectively.





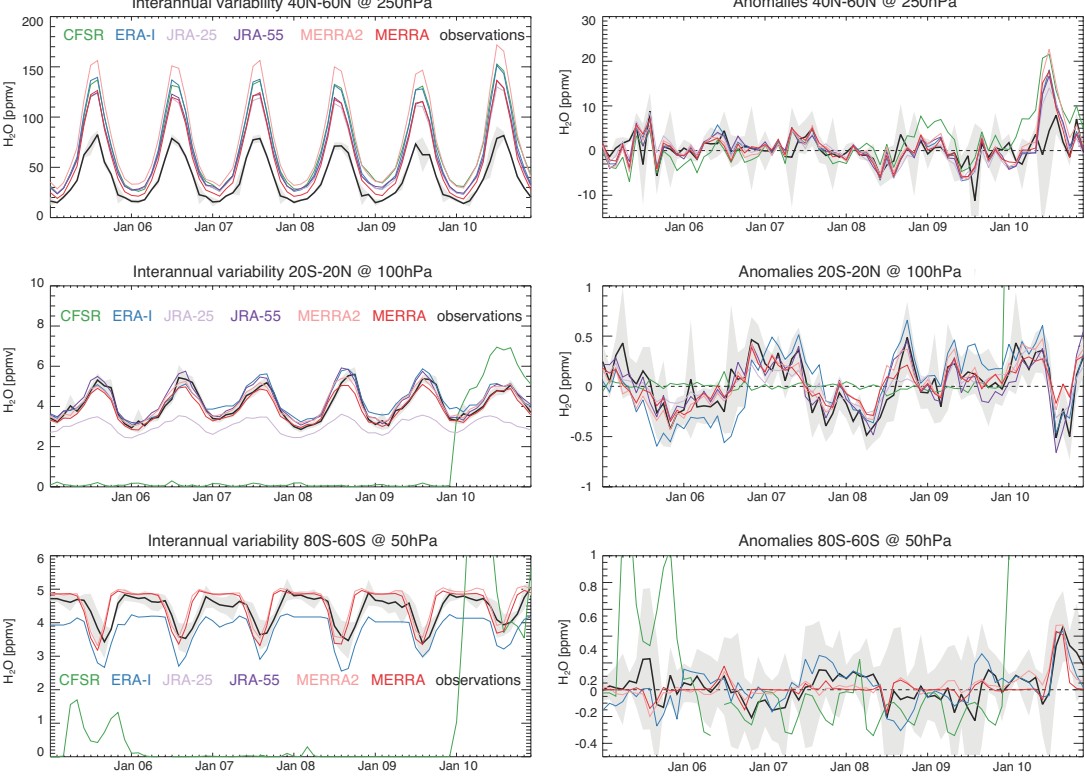

**Figure 14:** Interannual variability (left column) and deseasonalized anomalies (right column) for water vapour during 2005–2010 for the SPARC Data Initiative multi-instrument mean (SDI MIM, black) and the six reanalyses (coloured). Results are shown for three different pressure levels and latitude ranges (top to bottom: 50 hPa at 60–80°S, 100 hPa at 20°S–20°N, and 250 hPa at 40–60°N). Grey shading indicates observational uncertainty (±1-sigma) calculated as the standard deviation over all instruments and years available.



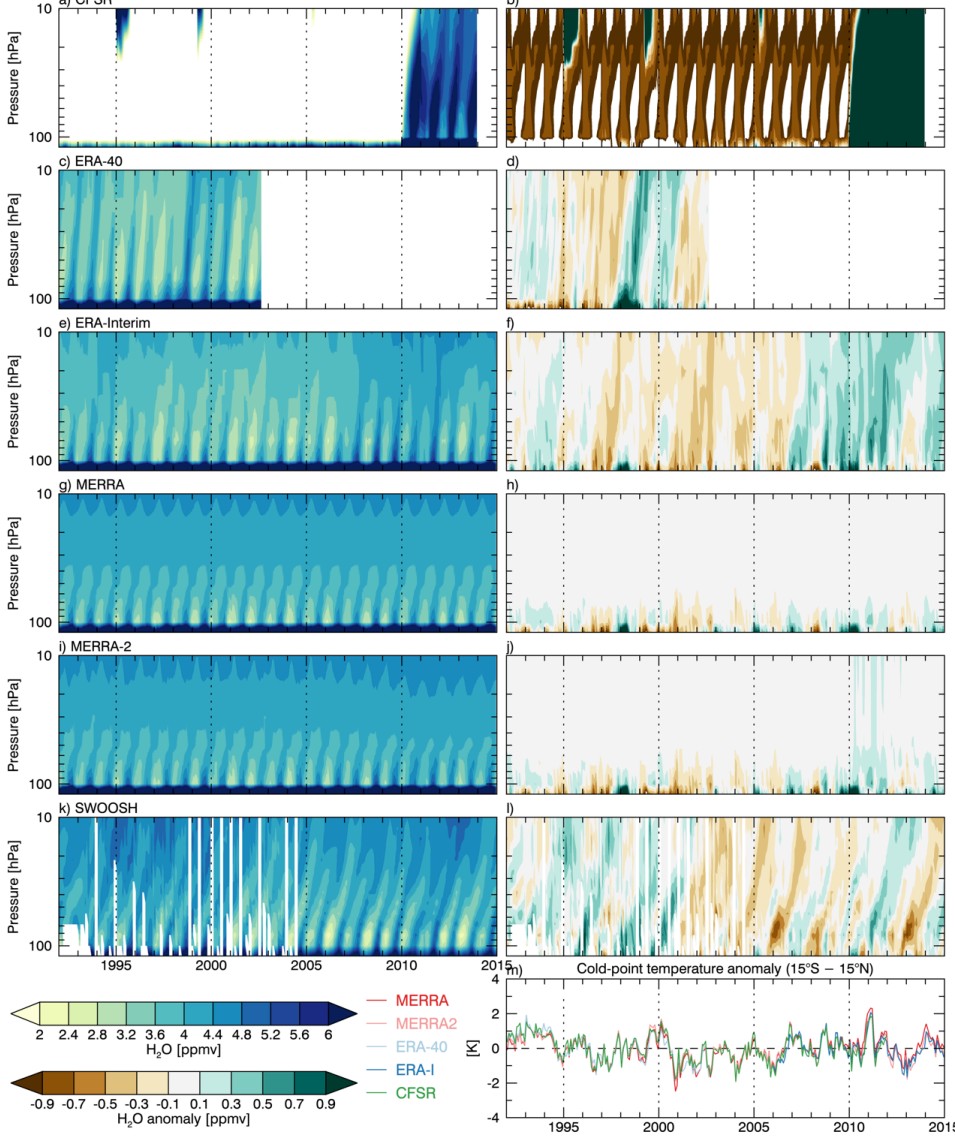

**Figure 15:** The tropical tape recorder signal as represented in reanalyses and the SWOOSH merged satellite product, defined as the height–time evolution of water vapour averaged over the 15°S–15°N tropical band. Both absolute values (left column) and anomalies relative to the mean water vapour seasonal cycle at each level (right column) are shown. Anomalies are computed separately for each data set. Monthly mean anomalies in tropical (15°S–15°N) cold-point tropopause temperatures calculated from 6-h data on the native vertical resolution of each reanalysis model are shown for context (m).