# Peer review of "Assessment of upper tropospheric and stratospheric water vapor and ozone in reanalyses as part of S-RIP"

_Atmospheric Chemistry and Physics, 2017_

## Referee Comment (RC1) · Anonymous Referee #1 · 4 Jul 2017

This paper is comprehensive and well written. It provides a lot of information on the ozone and water vapour fields in various state-of-the-art reanalyses, including quantification of their accuracy, usefulness of the datasets, and possible improvements. As such, I expect this paper to be useful to the atmospheric sciences community, and likely to be highly cited. I recommend publication in ACP subject to the authors paying attention to the specific comments below.

Specific comments

P. 3

[Figure]

L. 7: I suggest you indicate here what you will discuss in each section.

P. 4

L. 16: Do you need "notable"?

L. 27: It would be helpful to the reader to identify the old and updated forecast model and data assimilation system.

P. 10

L. 71: was -> were.

P. 27

L. 17: Maybe I am wrong, but I understood that there was a debate on the sign of trends in stratospheric water vapour during the late 1990s and early 2000s, with discrepancies between balloon and satellite measurements. Perhaps this has been resolved. Maybe the authors could mention this when they mention the work of Randel et al. (2006).

P. 41

Table 1: If you are using US spelling, it should be "analyzed". Same elsewhere.

P. 43

Fig. 1: What do the colours represent? Same for Fig. 2.

P. 45

Fig. 3: It would be helpful if the authors could identify in the caption what the red/blue colours indicate, e.g., positive/negative values. Same for other figures.

P. 47

Fig. 5: It would be helpful if the authors identified in the caption the colours referring to the reanalyses. Same for Fig. 6, 13, 14.

[Figure]

---

## Referee Comment (RC2) · Anonymous Referee #2 · 24 Jul 2017

This kind of paper is hard to review. It provides a summary of ozone and water vapour information in current reanalyses data sets, with an emphasis on the stratosphere. I guess the main conclusion of the paper was known before it was written: use the ozone and water vapour data with care and do not use for trend studies. However, it is nice to have some of the issues illustrated with figures in a comparative way. Therefore, there is no reason why the paper should not be published, however some explanations might require some clarifications. I will detail my questions below:

P3, l22: I do not know what this statement means. All systems try to model the microphysics of water as good as they can, jet the results differ, because small differences

[Figure]

in the treatment of water can have large effects?

P2, l30: This sentence is confusing. It tries to make two points in one sentence: Heating rate calculations and photochemistry. Which ozone is used when and where?

P6, l3: This relates to the comment regarding P2, l30. I guess a clear discussion in the beginning would be fine. Alternatively, a corresponding sentence for each system. Which ozone and water vapour is used in the radiation (heating rates) and what is done for the chemistry (actinic fluxes, if required).

P7, l9: See above.

P7, l34: was should read has.

P9, l6: I am not sure why the stratospheric temperature bias changes the humidity product. Ice clouds in Antarctica.

P11, l2: This sentence is not very clear. Which mean? What tendency? (For people in the know it will be clear, but . . .)

P11, l28: If I understand correctly, profiles (ozone and water vapour) are processed as described a few lines above on pressure levels and a common grid. However, TCO is calculated from model level data. Why not use the "ready made" products? Do some systems not provide their columns? How do you deal with orography?

P12/13: I appreciate that the authors would like to compare the reanalyses systems with another data source. However the data used is neither independent nor in a fundamental form. Instead, merged data sets are utilised. Nothing wrong with this, but presumably other equally valid products exist and I am not sure why the data sets mentioned have been chosen . . . (I am also not very happy with the use of multi-instrument means without an appreciate of the spread, when comparing to the reanalyses systems.)

Instead of focusing on the text, I will now briefly comment on a smal number of figures

(my assumption being, that readers will be most interested in the graphical presentation of system differences). Given the large range in water vapour products and the small number of systems that provide it, I will not comment on the water vapour related figures (the figures are a health warning in themselves):

Figure 3 and 4: I struggle to combine the information in both figures. For example JRA-25: In Figure 3 JRA-25 has a low bias with respect to SBUV everywhere. In Figure 4 JRA-25 has a positive bias from around 100 hPa to just below 10 hPa. Assuming that the largest column contribution stems from this region, I do not understand the consistency of the results. (Maybe I have over-read the explanation in the text . . .)

Figure 7: I find this figure hard to understand. Presumably, by using equivalent latitudes, the differences in more than one variable are highlighted. PV will have been derived from very different dynamical cores and afterwards ozone has been mapped to it. Therefore, differences will arise from more than one change in the assimilation system and how PV has been derived (e.g. treatment of temperature, dynamical variables and ozone itself, etc.). Therefore, I am not entirely sure what the message on a global scale is . . . apart from they all look different.

---

## Author Comment (AC1) · 22 Aug 2017

Below is our response to reviewer #1. Reviewer comments are in quotes, and our replies are inline below each reviewer comment

Review of Davis et al. "This paper is comprehensive and well written. It provides a lot of information on the ozone and water vapour fields in various state-of-the-art reanalyses, including quantification of their accuracy, usefulness of the datasets, and possible improvements. As such, I expect this paper to be useful to the atmospheric sciences community, and likely to be highly cited. I recommend publication in ACP subject to the authors paying attention to the specific comments below."

[Figure]

We thank the reviewer for their comments, and indeed hope that this work will be a useful resource for the community.

Specific comments

"P. 3, L. 7: I suggest you indicate here what you will discuss in each section."

Done, at the end of section 1.

"P. 4 L. 16: Do you need "notable"?"

We removed this word.

"L. 27: It would be helpful to the reader to identify the old and updated forecast model and data assimilation system."

We've added some information on the new and old systems to the sentence. Also, the details of the differences between CFSR and earlier NCEP reanalyses are discussed in Saha et al. 2010, and we've now made this more clear in the sentence.

"P. 10 L. 71: was -> were. "

Done

"P. 27 L. 17: Maybe I am wrong, but I understood that there was a debate on the sign of trends in stratospheric water vapour during the late 1990s and early 2000s, with discrepancies between balloon and satellite measurements. Perhaps this has been resolved. Maybe the authors could mention this when they mention the work of Randel et al. (2006)."

As the reviewer notes, there is a discrepancy between decadal trend estimates from balloon measurements and those from satellites. However, the Randel et al. 2006 paper and the text in question by the reviewer are referring to the drop in water vapor around the year 2000, and the measurements from both balloon and satellites are broadly in agreement on the existence of this drop. We have intentionally chosen to

not include reanalysis trends in WV or O3 in this paper.

"P. 41 Table 1: If you are using US spelling, it should be "analyzed". Same elsewhere."

Done. The entire document is now in US spelling.

"P. 43 Fig. 1: What do the colours represent? Same for Fig. 2."

The colors denote the reanalyses. We have made this clearer by coloring the reanalysis text labels with the corresponding color.

"P. 45 Fig. 3: It would be helpful if the authors could identify in the caption what the red/blue colours indicate, e.g., positive/negative values. Same for other figures."

Done

"P. 47 Fig. 5: It would be helpful if the authors identified in the caption the colours referring to the reanalyses. Same for Fig. 6, 13, 14."

The captions are already quite verbose, and we don't think it is necessary to do this since in all of these figures a legend is given showing the reanalyses and their colors.

---

## Author Comment (AC2) · 23 Aug 2017

Our response to the review is in line below, with responses given in bold:

This kind of paper is hard to review. It provides a summary of ozone and water vapour information in current reanalyses data sets, with an emphasis on the stratosphere. I guess the main conclusion of the paper was known before it was written: use the ozone and water vapour data with care and do not use for trend studies. However, it is nice to have some of the issues illustrated with figures in a comparative way. Therefore, there is no reason why the paper should not be published, however some explanations might require some clarifications. I will detail my questions below:

[Figure]

P3, l22: I do not know what this statement means. All systems try to model the micro-physics of water as good as they can, jet the results differ, because small differences in the treatment of water can have large effects?

**This sentence was confusing and has been re-worded. The point of the sentence was to state that SWV is not well constrained by observations because no obser-vations of SWV are assimilated, and that hence SWV is highly controlled by the differing physical representations of SWV. It now reads "Because stratospheric water vapour data are not directly assimilated, the concentration of water vapour in the stratosphere is highly variable amongst the reanalyses and is strongly af-fected by their representation of processes controlling it."**

P2, l30: This sentence is confusing. It tries to make two points in one sentence: Heating rate calculations and photochemistry. Which ozone is used when and where?

**(We assume the reviewer means P3 here, not P2) The point of this sentence was to note the large variety of ways ozone is represented in reanalyses. The information pertaining to the reviewers question is all summarized in Table 1. We realized that Table 1 was not referenced until later in the manuscript, so we've added reference to Table 1 here, and have also changed the wording to make our point more clear.**

P6, l3: This relates to the comment regarding P2, l30. I guess a clear discussion in the beginning would be fine. Alternatively, a corresponding sentence for each system. Which ozone and water vapour is used in the radiation (heating rates) and what is done for the chemistry (actinic fluxes, if required). P7, l9: See above.

**As noted above, this information is contained in Table 1. It is also discussed separately for each reanalysis.**

P7, l34: was should read has.

**Fixed**

P9, l6: I am not sure why the stratospheric temperature bias changes the humidity product. Ice clouds in Antarctica.

**In general, temperatures in the TTL affect how much water vapor enters the stratosphere. We have changed the wording here to make this point clear.**

P11, l2: This sentence is not very clear. Which mean? What tendency? (For people in the know it will be clear, but . . .)

**We agree with the reviewer that this sentence wasn't very clear. We've changed the sentence to be clearer about the potential impact of changes made in MERRA-2 on water vapor. The sentence now reads: "The main innovation in MERRA-2 that could impact water vapor is the introduction of additional global constraints that ensure continuity of water mass in the atmosphere (Takacs et al., 2016)."**

P11, l28: If I understand correctly, profiles (ozone and water vapour) are processed as described a few lines above on pressure levels and a common grid. However, TCO is calculated from model level data. Why not use the "ready made" products? Do some systems not provide their columns? How do you deal with orography?

**We misspoke in this paragraph. We computed the monthly means ourselves "from the 6-hourly TCO fields", not "from the 6 hourly model level data" as stated in the paper. We've re-worded this paragraph to make this clear. Orography is taken into account in using model level data because the lowest level is the surface, regardless of orographic height. As long as pressure is properly registered on the model levels (which it is in our JRA-25 calculations), using model level data is the most accurate way to compute total column ozone.**

P12/13: I appreciate that the authors would like to compare the reanalyses systems with another data source. However the data used is neither independent nor in a fundamental form. Instead, merged data sets are utilised. Nothing wrong with this, but

presumably other equally valid products exist and I am not sure why the data sets mentioned have been chosen . . . (I am also not very happy with the use of multi-instrument means without an appreciate of the spread, when comparing to the reanalyses systems.)

**We appreciate the reviewers concern that some of the comparisons in this paper are to assimilated (non-independent) observations. However, we note that this is only true for ozone, as all of our comparisons to water vapor data are to independent measurements. Also, for ozone we have carefully noted instances where the comparisons are not to independent data sources. We chose SPARC DI and SWOOSH data because we are familiar with these data sets. Other merged data sets (e.g., GOZCARDS) could be used, but based on outside work assessing the merged data sets, we have no reason to believe that using a different data set would alter the conclusions here (see, e.g., Tummon et al., ACP, 2015; Hubert et al., AMT, 2016; Harris et al., ACP, 2015). Regarding the spread around the multi instrument mean, the spread is shown in all of the line plots containing the MIM (Figs. 5,6,13,14).**

Instead of focusing on the text, I will now briefly comment on a smal number of figures (my assumption being, that readers will be most interested in the graphical presentation of system differences). Given the large range in water vapour products and the small number of systems that provide it, I will not comment on the water vapour related figures (the figures are a health warning in themselves): Figure 3 and 4: I struggle to combine the information in both figures. For example JRA- 25: In Figure 3 JRA-25 has a low bias with respect to SBUV everywhere. In Figure 4 JRA-25 has a positive bias from around 100 hPa to just below 10 hPa. Assuming that the largest column contribution stems from this region, I do not understand the consistency of the results. (Maybe I have over-read the explanation in the text)

**The reviewer has uncovered what turned out to be a major bug in the processing of the JRA-25 data. As noted in the manuscript, the JRA-25 total column ozone**

**data were processed directly from the 6-hourly model level data set. The verti-
cally resolved ozone came ultimately from the pressure level data supplied by
JMA and NCAR via CREATE-IP. It turns out that the pressure level data used the
incorrect hybrid level model coefficients when converting from model levels to
pressure levels, resulting in a downward "shift" of the entire ozone profile, as
seen in the old version of Figure 4. This was caused by the use of the model "in-
terface" hybrid coefficients, rather than the correct model "full level" coefficients
when converting from the model levels to pressure levels.**

**We have gone back to the original model level data to properly create the monthly
mean pressure level data set, and have updated all of the JRA-25 plots in this
paper that contain vertically resolved data. The affected figures that have been
updated are Figs. 4-6, 8, and 11 (the plots of TCO are correct). We have also
corrected the relevant discussion in the manuscript and added text explaining
our data processing for JRA-25 ozone.**

Figure 7: I find this figure hard to understand. Presumably, by using equivalent lati-
tudes, the differences in more than one variable are highlighted. PV will have been de-
rived from very different dynamical cores and afterwards ozone has been mapped to it.
Therefore, differences will arise from more than one change in the assimilation system
and how PV has been derived (e.g. treatment of temperature, dynamical variables and
ozone itself, etc.). Therefore, I am not entirely sure what the message on a global scale
is . . . apart from they all look different.

**The reviewer is correct that the plots showing ozone on isentropic-equivalent
latitude coordinates could be affected by differences in both the dynamical and
chemical (i.e., ozone) representation among the reanalyses. However, we believe
Figure 7 is valuable because it illustrates the sort of analysis that people do in
"research" studies, so we should know how derived quantities (such as ozone
on EqL-theta coordinates) compare among the reanalyses. Also, the figure quite
clearly illustrates some of the problems with ozone in the polar regions. The sim-**

**ilarity of the PV contours among the different reanalyses in this region suggests that the problem lies more in the representation of ozone than in the representation of vortex dynamics. We have added text in Section 4.5 to clarify these points and the importance of this figure.**